# Can Adversarial Training Be Manipulated By Non-Robust Features?

**Lue Tao**[1]*    **Lei Feng**[2,3]    **Hongxin Wei**[4]
**Jinfeng Yi**[5]    **Sheng-Jun Huang**[6]    **Songcan Chen**[6]†

[1]National Key Laboratory for Novel Software Technology, Nanjing University, Nanjing, China
[2]Chongqing University, Chongqing, China
[3]RIKEN Center for Advanced Intelligence Project, Japan
[4]Nanyang Technological University, Singapore
[5]JD AI Research, Beijing, China
[6]MIIT Key Laboratory of Pattern Analysis and Machine Intelligence,
Nanjing University of Aeronautics and Astronautics, Nanjing, China

## Abstract

Adversarial training, originally designed to resist test-time adversarial examples, has shown to be promising in mitigating *training-time availability attacks*. This defense ability, however, is challenged in this paper. We identify a novel threat model named *stability attack*, which aims to hinder *robust* availability by slightly manipulating the training data. Under this threat, we show that adversarial training using a conventional defense budget $\epsilon$ provably fails to provide test robustness in a simple statistical setting, where the non-robust features of the training data can be reinforced by $\epsilon$-bounded perturbation. Further, we analyze the necessity of enlarging the defense budget to counter stability attacks. Finally, comprehensive experiments demonstrate that stability attacks are harmful on benchmark datasets, and thus the adaptive defense is necessary to maintain robustness.[1]

## 1 Introduction

Robustness to input perturbations is crucial to machine learning deployment in various applications, such as spam filtering [13] and autonomous driving [6]. One of the most popular methods for improving test robustness is *adversarial training* [39, 1]. By augmenting the training data with $\epsilon$-bounded and on-the-fly crafted adversarial examples, adversarial training helps the learned model resist test-time perturbations [39].

On the other hand, machine learning systems are vulnerable to *training-time availability attacks* [3]. In particular, small perturbations applied into the training data (before training) suffice to degrade the overall test performance of naturally trained models [16, 28]. Fortunately, recent work has proven that adversarial training [39] is capable of mitigating this type of threat [64]. In other words, even if the training data is manipulated to maximize the test error, considerable accuracy on clean test data can still be achieved by adversarially trained models. However, previous work hardly inspects the test robustness of the models, which is what adversarial training was originally proposed for [23, 39]. This naturally raises the following question:

*Are the models adversarially trained on the manipulated data robust to test-time perturbations?*

---

*Work done when Lue Tao was a master's student at Nanjing University of Aeronautics and Astronautics.
†Corresponding author: Songcan Chen <s.chen@nuaa.edu.cn>.

[1]Our code is available at `https://github.com/TLMichael/Hypocritical-Perturbation`.

36th Conference on Neural Information Processing Systems (NeurIPS 2022).

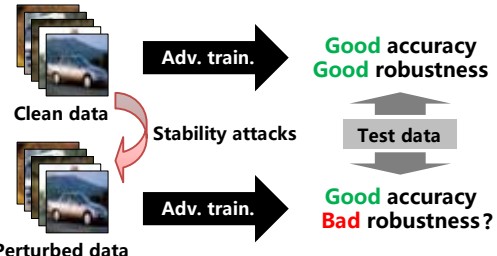

Figure 1: An illustration of stability attacks, where the training data is slightly perturbed to hinder adversarial training.

In this work, we show that conventional adversarial training may fail to provide test robustness when the training data is manipulated by an adversary, and thus an adaptive defense is necessary to resolve this issue. Our contributions are summarized as follows:

1. We introduce a novel threat model called *stability attack*, where an adversary aims to degrade the overall test robustness of adversarially trained models by slightly perturbing the training data. Figure 1 illustrates the threat of stability attacks.

2. We show that adversarial training using a conventional defense budget *provably* fails under stability attacks in a simple statistical setting. Specifically, a defense budget of $\epsilon$ will produce models that are *not* robust to $\epsilon$-bounded adversarial examples when the training data is hypocritically perturbed.

3. We unveil that the aforementioned vulnerability stems from the existence of the non-robust (predictive, yet brittle) features [29] in the original training data. When the non-robust features are reinforced by hypocritical perturbations, the conventional defense budget will be insufficient to offset the negative impact.

4. We further show that a defense budget of $2\epsilon$ is capable of resisting any stability attack for adversarial training, while the budget can be reduced to $\epsilon + \eta$ in a simple statistical setting, where $\eta$ is the magnitude of the non-robust features.

5. We demonstrate that stability attacks are harmful to conventional adversarial training on benchmark datasets. In addition, our empirical study suggests that enlarging the defense budget is essential for mitigating hypocritical perturbations.

To the best of our knowledge, this is the first work that studies the robustness of adversarial training against stability attacks. Both theoretical and empirical evidences show that the conventional defense budget $\epsilon$ is insufficient under the threat of $\epsilon$-bounded training-time perturbations. Our findings suggest that practitioners should consider a larger defense budget of no more than $2\epsilon$ (practically, about $1.5\epsilon \sim 1.75\epsilon$) to achieve a better $\epsilon$-robustness.

## 2 Threat Models

In this section, we formally introduce the threat model of stability attacks. We begin by revisiting the concepts of natural risk, adversarial risk, and delusive attacks. These concepts naturally give rise to our formulation of stability attacks.

### 2.1 Preliminaries

**Setup.** We consider a classification task with input-label pairs $(\boldsymbol{x}, y)$ from an underlying distribution $\mathcal{D}$ over $\mathcal{X} \times [k]$. The goal is to learn a (robust) classifier $f : \mathcal{X} \to [k]$ that predicts a label $y$ for a given input $\boldsymbol{x}$.

**Natural training (NT).** Most learning algorithms aim to maximize the generalization performance on unperturbed examples, i.e., natural accuracy. The goal is to minimize the natural risk defined as:

$$\mathcal{R}_{\text{nat}}(f) := \mathop{\mathbb{E}}_{(\boldsymbol{x},y)\sim\mathcal{D}} \left[ \mathcal{L}(f(\boldsymbol{x}), y) \right]. \tag{1}$$

Table 1: Comparisons between evasion attacks, delusive attacks, and stability attacks.

| Threat model | Training-time perturbation | Test-time perturbation | Learning scheme | Test performance |
|---|---|---|---|---|
| None | ✗ | ✗ | NT | Good |
| Evasion attacks [5, 60, 23, 39] | ✗ | ✓ | NT | Bad |
| | | | AT | Good |
| Delusive attacks [44, 16, 28, 64] | ✓ | ✗ | NT | Bad |
| | | | AT | Good |
| Stability attacks (this paper) | ✓ | ✓ | AT (conventional) | Bad |
| | | | AT (our improved) | Good |

**Adversarial training (AT).** Since the risk of adversarial examples (a.k.a. evasion attacks) was found to be unexpectedly high [5, 60], it has become increasingly important to defend the learner against the worst-case perturbations [23, 39]. In this context, the goal is to train a model that has low *adversarial risk* given a defense budget $\epsilon$:

$$\mathcal{R}_{\mathrm{adv}}(f) := \mathop{\mathbb{E}}_{(\boldsymbol{x}, y) \sim \mathcal{D}} \left[ \max_{\boldsymbol{\delta} \in \Delta} \mathcal{L}(f(\boldsymbol{x} + \boldsymbol{\delta}), y) \right], \tag{2}$$

where we choose $\Delta$ to be the set of $\epsilon$-bounded perturbations, i.e., $\Delta = \{\boldsymbol{\delta} \mid \|\boldsymbol{\delta}\| \leq \epsilon\}$. This choice is the most common one in the context of adversarial examples [67]. To simplify the notation, we refer to the robustness with respect to this set as $\epsilon$-robustness. It is worth noting that $\mathcal{R}_{\mathrm{adv}}(f) \geq \mathcal{R}_{\mathrm{nat}}(f)$ always holds for any $f$, and the equation holds when $\epsilon = 0$.

**Delusive attacks.** Delusive attacks, which belong to training-time availability attacks, aim to prevent the learner from producing an accurate model by manipulating the training data "imperceptibly" [44]. Concretely, the features of the training data can be perturbed, while the labels should remain correct [16, 42, 56, 28, 81, 15, 64, 18, 19]. This malicious task can be formalized into the following bi-level optimization problem:

$$\max_{\mathcal{P} \in \mathcal{S}} \mathop{\mathbb{E}}_{(\boldsymbol{x}, y) \sim \mathcal{D}} \left[ \mathcal{L}(f_{\mathcal{P}}(\boldsymbol{x}), y) \right]$$
$$\text{s.t. } f_{\mathcal{P}} \in \arg\min_{f} \sum_{(\boldsymbol{x}_i, y_i) \in \mathcal{T}} \left[ \mathcal{L}(f(\boldsymbol{x}_i + \boldsymbol{p}_i), y_i) \right]. \tag{3}$$

Here, the adversary aims to maximize the natural risk of the model $f_{\mathcal{P}}$ (that is trained on the manipulated training set) by applying the generated perturbations $\mathcal{P} = \{\boldsymbol{p}_i\}_{i=1}^n$ into the original training set $\mathcal{T} = \{(\boldsymbol{x}_i, y_i)\}_{i=1}^n$. The commonly used feasible region is $\mathcal{S} = \{\{\boldsymbol{p}_i\}_{i=1}^n \mid \|\boldsymbol{p}_i\| \leq \epsilon\}$.

Generally, solving Equation (3) is computationally prohibitive for neural networks [64, 18]. Thus, various heuristic methods are proposed to achieve the goal. Among them, a representative method is the *hypocritical perturbation* [63, 64], crafted as follows:

$$\min_{\|\boldsymbol{p}_i\| \leq \epsilon} \mathcal{L}(f_{\mathrm{craft}}(\boldsymbol{x}_i + \boldsymbol{p}_i), y_i), \tag{4}$$

where $f_{\mathrm{craft}}$ is called the *crafting model*, pre-trained before generating poisons. Tao et al. [64] simply adopted a naturally trained classifier as the crafting model, while Huang et al. [28] proposed a min-min bi-level optimization process to pre-train the crafting model. Fu et al. [19] further built their crafting model via a min-min-max three-level optimization process, and generated their poisons by replacing Equation (4) with a min-max bi-level objective.

Another representative method of delusive attacks is the *adversarial perturbation*, crafted by solving

$$\max_{\|\boldsymbol{p}_i\| \leq \epsilon} \mathcal{L}(f_{\mathrm{craft}}(\boldsymbol{x}_i + \boldsymbol{p}_i), y_i). \tag{5}$$

Tao et al. [64] and Fowl et al. [18] both found that applying the adversarial perturbation to the training data is very effective at compromising naturally trained models. However, adversarial training has proven to be promising in defending against various delusive attacks [64].

## 2.2 Stability Attacks

In contrast to delusive attacks that aim at increasing the natural risk, stability attacks attempt to maximize the adversarial risk of the learner by slightly perturbing the training data:

$$\max_{\mathcal{P} \in \mathcal{S}} \mathbb{E}_{(\boldsymbol{x}, y) \sim \mathcal{D}} \left[ \max_{\boldsymbol{\delta} \in \Delta} \mathcal{L}(f_{\mathcal{P}}(\boldsymbol{x} + \boldsymbol{\delta}), y) \right], \tag{6}$$

where $f_{\mathcal{P}}$ denotes the victim model, which is naturally or adversarially trained on the perturbed data. In other words, stability attacks seek to hinder the *robust* availability of the training data. Table 1 shows the comparisons among different threat models.

The goal of stability attacks can be immediately achieved for naturally trained models, since they have already incurred high adversarial risk, even if the training data is clean [60]. To ease the problem of high adversarial risk, adversarial training has been widely used to improve model's adversarial robustness [24, 12]. Hence, the main goal of stability attacks becomes to compromise the test robustness of adversarially trained models.

Note that Equation (6) is a multi-level optimization problem that is not easy to solve, our next question is how to conduct effective stability attacks against adversarial training. In the following sections, we introduce an effective stability attack method and analyze the cost of resisting it.

**Remark 1.** This work focuses on adding bounded pertubrations as small as possible. We mostly assume that the adversary has full control of training data (instead of changing a few) by following previous works [16, 28, 64, 17–19]. This is a realistic assumption [16, 18]. For instance, in some applications an organization may agree to release some internal data for peer assessment, while preventing competitors from easily building a model with high test robustness; this can be achieved by perturbing the entire dataset via stability attacks before releasing. Moreover, this assumption enables a worst-case analysis of the robustness of adversarial training, which may facilitate important theoretical implications.

## 3 How to Manipulate Adversarial Training

Previous work suggests that adversarial training could defend against both evasion attacks and delusive attacks [39, 64]. However, in this paper, we show that adversarial training using a conventional defense budget $\epsilon$ may not be sufficient to provide $\epsilon$-robustness when confronted with stability attacks. In particular, we present a simple theoretical model where the conventional defense scheme provably fails when the training data is hypocritically perturbed.

**The binary classification task.** The data model is largely based on the setting proposed by Tsipras et al. [67], which draws a distinction between *robust features* and *non-robust features*. Specifically, it consists of input-label pairs $(\boldsymbol{x}, y)$ sampled from a Gaussian mixture distribution $\mathcal{D}$ as follows:

$$y \overset{u.a.r}{\sim} \{-1, +1\}, \quad x_1 \sim \mathcal{N}(y, \sigma^2), \quad x_2, \ldots, x_{d+1} \overset{i.i.d}{\sim} \mathcal{N}(\eta y, \sigma^2), \tag{7}$$

where $\eta$ is much smaller than 1 (i.e., $0 < \eta \ll 1$). Hence, samples from $\mathcal{D}$ consist of a robust feature ($x_1$) that is *strongly* correlated with the label, and $d$ non-robust features ($x_2, \ldots, x_{d+1}$) that are *very weakly* correlated with it. Typically, an adversary can manipulate a large number of non-robust features - e.g. $d = \Theta(1/\eta^2)$ will suffice.

Before introducing the way to hinder robust availability, we briefly illustrate the success of adversarial training when the training data is unperturbed.

**Natural and robust classifiers.** For standard classification, we consider a natural classifier:

$$f_{\text{nat}}(\boldsymbol{x}) := \text{sign}(\boldsymbol{w}_{\text{nat}}^{\top} \boldsymbol{x}), \text{ where } \boldsymbol{w}_{\text{nat}} := [1, \eta, \ldots, \eta], \tag{8}$$

which is a minimizer of the natural risk (1) with 0-1 loss on the data (7), i.e., the Bayes optimal classifier. However, in the adversarial setting, this natural classifier is quite brittle. Thus, it is imperative to obtain a robust classifier:

$$f_{\text{rob}}(\boldsymbol{x}) := \text{sign}(\boldsymbol{w}_{\text{rob}}^{\top} \boldsymbol{x}), \text{ where } \boldsymbol{w}_{\text{rob}} := [1, 0, \ldots, 0], \tag{9}$$

which relies only on the robust feature $x_1$.

**Illustration of adversarial accuracy.**    In the adversarial setting, an adversary that is only allowed to perturb each feature by a moderate $\epsilon$ can effectively subvert the natural classifier Tsipras et al. [67]. In particular, if $\epsilon = 2\eta$, an adversary can essentially force each non-robust feature to be *anti*-correlated with the correct label. The following proposition, proved in Appendix C.1, gives the adversarial accuracies of the natural classifier $f_{\text{nat}}$ (8) and the robust classifier $f_{\text{rob}}$ (9).

**Proposition 1.** *Let $\epsilon = 2\eta$ and denote by $\mathcal{A}_{\text{adv}}(f)$ the adversarial accuracy, i.e., the probability of a classifier correctly predicting $y$ on the data (7) under $\ell_\infty$ perturbations. Then, we have*

$$\mathcal{A}_{\text{adv}}(f_{\text{nat}}) \leq \Pr\left\{ \mathcal{N}(0,1) < \frac{1 - d\eta^2}{\sigma\sqrt{1 + d\eta^2}} \right\}, \quad \mathcal{A}_{\text{adv}}(f_{\text{rob}}) = \Pr\left\{ \mathcal{N}(0,1) < \frac{1 - 2\eta}{\sigma} \right\}.$$

Proposition 1 implies that the adversarial accuracy of the natural classifier is $< 50\%$ when $d \geq 1/\eta^2$. Even worse, when $\sigma \leq 1/3$ and $d \geq 3/\eta^2$, the adversarial accuracy of the natural classifier (8) is always lower than $1\%$. In contrast, the robust classifier (9) yields a much higher adversarial accuracy (always $> 50\%$); when $\sigma \leq (1 - 2\eta)/3$, its adversarial accuracy will be higher than $99\%$.

## 3.1   Hypocritical Features Are Harmful

The results above reveal the advantages of robust classifiers over natural classifiers. Note that such a robust classifier can be obtained by adversarial training on the original data (7). However, this defense effect may not hold when the adversary is allowed to perturb the training data.

We show this by analyzing two representative perturbations: the *adversarial perturbation* [64, 18] and the *hypocritical perturbation* [63, 64]. When applied into the training data, both perturbations are effective as delusive attacks for naturally trained models. In the following, we show that the former is harmless: adversarial training using a defense budget $\epsilon$ on the adversarially perturbed data can still provide test robustness. In contrast, the latter is harmful: we find that the same defense budget can only produce non-robust classifiers when the training data is hypocritically perturbed.

**A harmless case.**    Consider an adversary who is capable of perturbing the training data by an attack budget $\epsilon$. The adversary may choose to shift each feature towards $-y$. Hence, the learner would see input-label pairs $(\boldsymbol{x}, y)$ sampled i.i.d. from a training distribution $\mathcal{T}_{\text{adv}}$ as follows:

$$y \overset{u.a.r}{\sim} \{-1, +1\}, \quad x_1 \sim \mathcal{N}((1 - \epsilon)y, \sigma^2), \quad x_2, \ldots, x_{d+1} \overset{i.i.d}{\sim} \mathcal{N}((\eta - \epsilon)y, \sigma^2), \tag{10}$$

where each feature of the samples from $\mathcal{T}_{\text{adv}}$ is adversarially perturbed by a moderate $\epsilon$. While these samples are deviate significantly from the original distribution $\mathcal{D}$ (7), adversarial training on them using a defense budget $\epsilon$ is still able to neutralize the non-robust features. Formally, in Appendix C.2 we prove the following theorem.

**Theorem 1** (Adversarial perturbation is harmless). *Assume that the adversarial perturbation in the training data $\mathcal{T}_{adv}$ (10) is moderate such that $\eta/2 \leq \epsilon < 1/2$. Then, the optimal linear $\ell_\infty$-robust classifier obtained by minimizing the adversarial risk on $\mathcal{T}_{adv}$ with a defense budget $\epsilon$ is equivalent to the robust classifier (9).*

This theorem indicates that the adversarial perturbation is harmless: $\epsilon$-robustness can still be obtained by adversarial training on such perturbed training data.

**A harmful case.**    However, this defense effect can be completely broken by the hypocritical perturbation. That is, the adversary can instead shift each feature towards $y$. Hence, the learner would see input-label pairs $(\boldsymbol{x}, y)$ sampled i.i.d. from a training distribution $\mathcal{T}_{\text{hyp}}$ as follows[2]:

$$y \overset{u.a.r}{\sim} \{-1, +1\}, \quad x_1 \sim \mathcal{N}((1 + \epsilon)y, \sigma^2), \quad x_2, \ldots, x_{d+1} \overset{i.i.d}{\sim} \mathcal{N}((\eta + \epsilon)y, \sigma^2), \tag{11}$$

where each feature of the samples from $\mathcal{T}_{\text{hyp}}$ is reinforced by a magnitude of $\epsilon$. While these samples become more separable, adversarial training on them using the same defense budget will fail to neutralize the hypocritically perturbed features. Consequently, the resulting classifiers will inevitably have low adversarial accuracy. We make this formal in the following theorem proved in Appendix C.3.

---

[2]To see how this relates to the hypocritical perturbation (4), let us consider the logistic loss $\mathcal{L}(f(\boldsymbol{x}), y) = \log(1 + \exp(-yf(\boldsymbol{x})))$, and use the natural classifier (8) as the crafting model. Then, the problem (4) has a closed-form solution $\boldsymbol{p}^* = y\epsilon \cdot \text{sign}(\boldsymbol{w}_{\text{nat}}) = [y\epsilon, \ldots, y\epsilon]$. Applying $\boldsymbol{p}^*$ to each $\boldsymbol{x}$ yields the distribution $\mathcal{T}_{\text{hyp}}$.

**Theorem 2** (Hypocritical perturbation is harmful). *The optimal linear $\ell_\infty$-robust classifier obtained by minimizing the adversarial risk on the perturbed data $\mathcal{T}_{hyp}$ (11) with a defense budget $\epsilon$ is equivalent to the natural classifier (8).*

This theorem implies that the conventional defense scheme can only produce non-robust classifiers, whose adversarial accuracy is as low as that of the natural classifier (8). That is saying, if $\epsilon = 2\eta$, $\sigma \leq 1/3$ and $d \geq 3/\eta^2$, the classifiers cannot get adversarial accuracy better than $1\%$.

**Implications.** As it turns out, the seemingly beneficial features in $\mathcal{T}_{hyp}$ (11) are actually hypocritical. Therefore, the adversary is highly motivated to hide such hypocritical features in the training data, intending to cajole an innocent learner into relying on the non-robust features. Intriguingly, we notice that the natural classifier (8) (i.e., the crafting model used to derive the distribution $\mathcal{T}_{hyp}$) actually has $\eta$-robustness. This is essentially because the non-robust features in the data (7) can resist small-magnitude perturbations by design. This motivates us to use "slightly robust" classifiers as the crafting model in practice. Indeed, our experimental results show that training the crafting model with $0.25\epsilon$-robustness performs the best for conducting stability attacks. This is different from the previous works [64, 18] that use naturally trained models as the crafting model for poisoning.

## 4 The Necessity of Large Defense Budget

We have shown that the hypocritical perturbation is harmful to the conventional adversarial training scheme. Fortunately, it is possible to strengthen the defense by using a larger defense budget, while the crux of the matter is how large the budget is needed.

We find that the minimum value of the defense budget for a successful defense depends on the specific data distribution. Let us first consider the hypocritical data in $\mathcal{T}_{hyp}$ (11). In this case, we show that a larger defense budget is necessary in the following theorem proved in Appendix C.4.

**Theorem 3** ($\epsilon + \eta$ is necessary). *The optimal linear $\ell_\infty$-robust classifier obtained by minimizing the adversarial risk on the perturbed data $\mathcal{T}_{hyp}$ (11) with a defense budget $\epsilon + \eta$ is equivalent to the robust classifier (9). Moreover, any defense budget lower than $\epsilon + \eta$ will yield classifiers that still rely on all the non-robust features.*

This theorem implies that, in the case of the mixture Gaussian distribution under the threat of $\epsilon$-bounded hypocritical perturbations, the learner needs a slightly larger defense budget $\epsilon + \eta$ to ensure $\epsilon$-robustness.

While it is challenging to analyze the minimum value of the defense budget in the general case, the following theorem provides an upper bound of the budget.

**Theorem 4** (General case). *For any data distribution and any adversary with an attack budget $\epsilon$, training models to minimize the adversarial risk with a defense budget $2\epsilon$ on the perturbed data is sufficient to ensure $\epsilon$-robustness.*

The proof of Theorem 4 is deferred in Appendix C.5. It implies that a defense budget twice to the attack budget should be safe enough under the threat of stability attacks. Theorem 3 also suggests that the minimum budget might be much smaller than $2\epsilon$, and it depends on the specific attack methods and data distributions. In the following section, we empirically search for an appropriate defense budget on real-world datasets.

## 5 Experiments

In this section, we conduct comprehensive experiments to demonstrate the effectiveness of the hypocritical perturbation as stability attacks on popular benchmark datasets and the necessity of an adaptive defense for better robustness.

We conduct stability attacks by applying hypocritical perturbations into the training set. We focus on an $\ell_\infty$ adversary with an *attack budget* $\epsilon_a = 8/255$ by following [28, 81, 64, 18]. Our crafting model is adversarially trained with a *crafting budget* $\epsilon_c = 2/255$ for 10 epochs before generating perturbations. Unless otherwise specified, we use ResNet-18 [26] as the default architecture for both the crafting model and the learning model. For adversarial training, we mainly follow the settings in

Table 2: Test robustness (%) of PGD-AT using a defense budget $\epsilon_d = 8/255$ on CIFAR-10.

| Attack | Natural | FGSM | PGD-20 | PGD-100 | $CW_\infty$ | AutoAttack |
|---|---|---|---|---|---|---|
| None (clean) | 82.17 | 56.63 | 50.63 | 50.35 | 49.37 | 46.99 |
| DeepConfuse [16] | 81.25 | 54.14 | 48.25 | 48.02 | 47.34 | 44.79 |
| Unlearnable Examples [28] | 83.67 | 57.51 | 50.74 | 50.31 | 49.81 | 47.25 |
| NTGA [81] | 82.99 | 55.71 | 49.17 | 48.82 | 47.96 | 45.36 |
| Adversarial Poisoning [18] | **77.35** | 53.93 | 49.95 | 49.76 | 48.35 | 46.13 |
| Hypocritical Perturbation (ours) | 88.07 | **47.93** | **37.61** | **36.96** | **38.58** | **35.44** |

Table 3: Test robustness (%) of PGD-AT using a defense budget $\epsilon_d = 8/255$ across different datasets.

| Dataset | Attack | Natural | FGSM | PGD-20 | PGD-100 | $CW_\infty$ | AutoAttack |
|---|---|---|---|---|---|---|---|
| | None | 93.95 | 71.83 | 57.15 | 56.02 | 54.93 | 50.50 |
| SVHN | Adv. | **87.50** | **56.12** | 46.71 | 46.32 | 45.70 | 42.48 |
| | Hyp. | 96.06 | 59.41 | **38.17** | **37.29** | **40.54** | **35.43** |
| | None | 56.15 | 31.50 | 28.38 | 28.28 | 26.53 | 24.30 |
| CIFAR-100 | Adv. | **52.14** | 28.59 | 26.19 | 26.09 | 24.36 | 22.71 |
| | Hyp. | 62.22 | **26.38** | **21.51** | **21.13** | **21.13** | **18.74** |
| | None | **49.34** | 25.67 | 22.99 | 22.86 | 20.67 | 18.54 |
| Tiny-ImageNet | Adv. | 49.52 | 22.93 | 20.01 | 19.91 | 18.75 | 16.83 |
| | Hyp. | 55.92 | **20.21** | **15.61** | **15.26** | **14.99** | **12.53** |

Table 4: Test robustness (%) of PGD-AT using a defense budget $\epsilon_d = 8/255$ on CIFAR-10 across different architectures. Test robustness is evaluated by PGD-20. Values in parenthesis denote the accuracy on natural test data.

| Attack | VGG-16 | GoogLeNet | DenseNet-121 | MobileNetV2 | WideResNet-28-10 |
|---|---|---|---|---|---|
| None | 47.37 (77.15) | 50.67 (83.03) | 49.92 (80.08) | 48.51 (80.83) | 53.91 (85.81) |
| Adv. | 44.70 (73.24) | 47.72 (79.34) | 48.00 (78.17) | 45.90 (74.61) | 51.01 (82.43) |
| Hyp. | **34.34** (87.20) | **37.03** (87.61) | **37.58** (88.04) | **35.58** (87.04) | **41.07** (89.14) |

previous studies [83, 70, 50]. By convention, the *defense budget* is equal to the attack budget, i.e., $\epsilon_d = 8/255$. More details on experimental settings are provided in Appendix D.

## 5.1 Benchmarking (Non-)Robustness

**Attack evaluation.** We compare our crafted hypocritical perturbation to existing methods, which were originally proposed as delusive attacks, including DeepConfuse (which builds an adversarial auto-encoder to generate their perturbations) [16], Unlearnable Examples (which use a min-min bi-level optimization process to pre-train their crafting model) [28], NTGA (which adopts neural tangent kernels as its crafting model) [81], and Adversarial Poisoning (whose crafting model is simply a naturally trained classifier) [18]. It is noteworthy that none of these previous works evaluated the test robustness of their poisoned models.

Results using ResNet-18 on CIFAR-10 are summarized in Table 2. "Natural" denotes the accuracy on natural test data. Various test-time adversarial attacks are used to evaluate test robustness, including FGSM, PGD-20/100, $CW_\infty$ ($\ell_\infty$ version of CW loss [9] optimized by PGD-100), and AutoAttack (a reliable evaluation metric via an ensemble of diverse attacks [11]). We observe that the hypocritical perturbation widely outperforms previous training-time perturbations in degrading the test robustness of PGD-AT [39]. This demonstrates that stability attacks are indeed harmful to the conventional defense scheme. We note that our method increases the natural accuracy. This is reasonable, since our analysis in Section 3.1 has implied that the hypocritical perturbation can increase model reliance on the non-robust features, which are predictive but brittle [29].

Moreover, we evaluate the hypocritical perturbation on other benchmark datasets including SVHN, CIFAR-100, and Tiny-ImageNet. Both the crafting model and the victim model use the ResNet-18

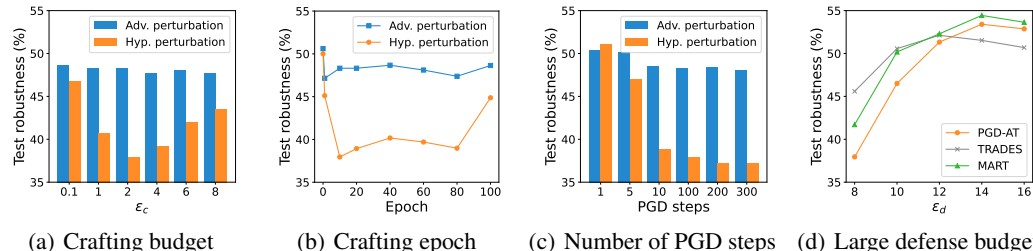

| | (a) Crafting budget | (b) Crafting epoch | (c) Number of PGD steps | (d) Large defense budget |

Figure 2: The ablation study experiments on CIFAR-10. Test robustness (%) is evaluated by PGD-20.

Table 5: Test robustness (%) of various adaptive defenses on the hypocritically perturbed CIFAR-10.

| Defense | Natural | FGSM | PGD-20 | PGD-100 | CW$_\infty$ | AutoAttack |
|---|---|---|---|---|---|---|
| PGD-AT ($\epsilon_d = 8/255$) | 88.07 | 47.93 | 37.61 | 36.96 | 38.58 | 35.44 |
| + Random Noise | 87.62 | 47.46 | 38.35 | 37.90 | 39.07 | 36.25 |
| + Gaussian Smoothing | 83.95 | 50.96 | 42.80 | 42.34 | 42.41 | 40.07 |
| + Cutout | **88.26** | 49.23 | 39.77 | 39.25 | 40.38 | 37.61 |
| + AutoAugment | 86.24 | 48.87 | 40.19 | 39.65 | 37.66 | 35.07 |
| PGD-AT ($\epsilon_d = 14/255$) | 80.00 | 56.86 | 52.92 | 52.83 | **50.36** | **48.63** |
| TRADES ($\epsilon_d = 12/255$) | 79.63 | 55.73 | 51.77 | 51.63 | 48.68 | 47.83 |
| MART ($\epsilon_d = 14/255$) | 77.29 | **57.10** | **53.82** | **53.71** | 49.03 | 47.67 |

Table 6: Test robustness (%) of PGD-AT by adjusting the amount of clean data included in the manipulated CIFAR-10. Test robustness (%) is evaluated by PGD-20. Values in parenthesis denote the accuracy on natural test data.

| Attack\Clean proportion | 0.1 | 0.2 | 0.4 | 0.6 | 0.8 |
|---|---|---|---|---|---|
| None (clean subset) | 30.65 (63.90) | 37.99 (70.99) | 44.95 (77.11) | 47.17 (80.33) | 49.78 (81.60) |
| Adversarial Perturbation | 48.33 (77.71) | 48.23 (76.94) | 49.68 (78.54) | 50.15 (82.46) | 51.21 (82.03) |
| Hypocritical Perturbation | **41.51** (87.49) | **43.66** (88.30) | **46.98** (86.46) | **49.20** (85.29) | **50.56** (82.72) |

architecture. Results are summarized in Table 3. "Hyp." denotes the hypocritical perturbation generated by our crafting model. As a comparison, we also evaluate the adversarial perturbation generated using the same crafting model (denoted as "Adv."). Again, the results show that the hypocritical perturbations are more threatening than the adversarial perturbations to standard adversarial training. This phenomenon is consistent with our analytical results in Section 3.1.

Besides, we find that the hypocritical perturbation can transfer well from ResNet-18 to other architectures, successfully degrading the test robustness of a wide variety of popular architectures including VGG-16 [57], GoogLeNet [61], DenseNet-121 [27], MobileNetV2 [52], and WideResNet-28-10 [82], as shown in Table 4. Note that this is a completely black-box setting where the attacker has no knowledge of the victim model's initialization, architecture, learning rate scheduler, etc.

**Adaptive defense.** To prevent the harm of stability attacks, our analysis in Section 4 suggests that a larger defense budget would be helpful. We find that this is indeed the case on CIFAR-10. As shown in Table 5, a large defense budget $\epsilon_d = 14/255$ for PGD-AT performs significantly better than the conventional defense budget $\epsilon_d = 8/255$. We also combine several data augmentations with PGD-AT as defenses by following Fowl et al. [18]. The results show that they are beneficial, while their improvements are inferior to PGD-AT with $\epsilon_d = 14/255$. In addition, we adopt other adversarial training variants including TRADES [83] and MART [70] to defend against the hypocritical perturbation, and find that they achieve comparable defense effects with large defense budgets.

Finally, we note that the adaptive defense has several limitations: *i)* robust accuracy is improved at the cost of natural accuracy; *ii)* finding an appropriate defense budget is time-consuming for adversarial training; *iii)* adversarial training with large budgets may lead to learning obstacles such as inherent large sample complexity [53]. We leave the detailed study of these questions as future work.

## 5.2 Ablation Studies

In this part, we conduct a set of experiments to provide an empirical understanding of the proposed attack. We train ResNet-18 using PGD-AT on CIFAR-10 by following the same settings described in Appendix D unless otherwise specified.

**Analysis on the crafting method.** Different from previous work, we use "slightly robust" classifiers as our crafting model. Figure 2(a) shows that this technique greatly improves the potency of the hypocritical perturbation, where the crafting budget $\epsilon_c = 2/255$ performs best in degrading test robustness. We also observe that training the crafting model for 10~80 epochs works well in Figure 2(b), and that optimizing the crafted perturbations over 100 steps performs well in Figure 2(c). Finally, we note that Fowl et al. [18] also tried to use adversarially trained models as the crafting model, but they failed to produce an effective attack in this way. This is mainly because they adopted adversarial perturbations as poisons, which, as we observed, are inferior in degrading test performance.

**Ablation on defense budget.** As discussed in Section 4, we are motivated to find the appropriate defense budget $\epsilon_d$ in the range $[\epsilon \sim 2\epsilon]$. Figure 2(d) shows that the optimal defense budgets against the hypocritical perturbation are $14/255$, $12/255$, and $14/255$ for PGD-AT, TRADES, and MART, respectively. We also observe that all these adversarial training variants are inferior when using the conventional defense budget $8/255$.

**Less data.** We follow Fowl et al. [18] to test the effectiveness of attacks by varying the proportion of clean data and perturbed data. Attacks are then considered effective if they cannot significantly increase performance over training on the clean subset alone. As shown in Table 6, the proposed attack often degrades the test robustness below what one would achieve using full clean dataset. More importantly, the hypocritical perturbations are consistently more harmful than the adversarial perturbations. This again verifies the superiority of hypocritical perturbations as stability attacks.

**Effect on natural training.** As a sanity check, we include the test accuracy of naturally trained models on CIFAR-10 in Table 7. It shows that without adversarial training, the test robustness of the models becomes very poor—all models only have $0\%$ accuracy under PGD-20 attack. Thus, the goal of stability attacks is immediately achieved. On the other hand, We find that our method degrades the test accuracy from $94.23\%$ to $75.92\%$, though this is not the main focus of

Table 7: Test robustness (%) of natural training on CIFAR-10.

| Attack | Natural | PGD-20 |
|---|---|---|
| None (clean) | 94.23 | 0.00 |
| DeepConfuse [16] | 17.22 | 0.00 |
| Unlearnable Examples [28] | 22.72 | 0.00 |
| NTGA [81] | 11.15 | 0.00 |
| Adversarial Poisoning [18] | **8.60** | 0.00 |
| Hypocritical Perturbation (ours) | 75.92 | 0.00 |

this work. We also observe that Adversarial Poisoning [18] is the most effective method in degrading the test accuracy of naturally trained models. This observation is consistent with Fowl et al. [18].

## 6 Related Work

**Adversarial training.** The presence of non-robust features has been demonstrated on popular benchmark datasets [29, 31], which naturally leads to model vulnerability to adversarial examples [67, 58]. To improve test robustness against adversarial examples, adversarial training methods have been developed [23, 39, 73, 83, 65, 46, 76, 86, 62, 69]. Usually, adversarial training using a defense budget $\epsilon$ is expected to improve model robustness against $\epsilon$-bounded adversarial examples. Thus, to break this defense, a direct way is to enlarge the typical $\epsilon$-ball used to constrain the attack; however, this may risk changing the true label [8, 66]. In this work, we aim to show that it is possible to achieve this by slightly perturbing the training data without enlarging the $\epsilon$-ball.

**Data poisoning.** Data poisoning attacks, which manipulate training data to cause the resulting models to fail during inference [3], can be divided into *availability attacks* (to degrade overall test performance) [4, 77, 41, 47, 18] and *integrity attacks* (to cause specific misclassifications) [32, 10, 55, 87, 21, 54]. While the stability attacks studied in this work may be reminiscent of *backdoor attacks* [10], we note that they share several key differences. First, stability attacks aim to hinder adversarial training with well-defined $\epsilon$-robustness, while backdoor attacks mainly focus on

embedding malicious behaviors (that can be invoked by pre-specified triggers) into naturally trained models [22, 51, 68]. Second, stability attacks only perturb the inputs slightly, while most works on backdoor attacks require mislabeling [25, 37, 45, 36, 74]. Thus, backdoor defenses [7, 75, 35] might not be directly applied to resist stability attacks.

Additional related works are discussed in Appendix A.

## 7 Conclusion

In this work, we establish a framework to study the robustness of adversarial training against stability attacks. We unveil the threat of stability attacks—small hypocritical perturbations applied into the training data suffice to hinder conventional adversarial training. The conventional defense budget $\epsilon$ is insufficient under the threat. To resist it, we suggest a larger defense budget of no more than $2\epsilon$. Our theoretical analysis explains why hypocritical perturbations are effective as stability attacks— they can mislead the learner by reinforcing the non-robust features. Experiments demonstrate that hypocritical perturbations are harmful to conventional adversarial training on benchmark datasets, and enlarging the defense budget is essential for mitigating hypocritical perturbations. Future work includes relaxing the assumption that the adversary perturbs the entire training set and designing more effective stability attacks against adversarial training.

## Acknowledgments and Disclosure of Funding

This work was supported by the National Natural Science Foundation of China (Grant No. 61732006, 62076124, 62076128, 62106028), the National Key R&D Program of China (2020AAA0107000). Lei Feng was also supported by Chongqing Overseas Chinese Entrepreneurship, Innovation Support Program and CAAI-Huawei MindSpore Open Fund. The authors wish to thank the anonymous reviewers for their helpful comments and suggestions.

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
