# Supplementary Material: Can Adversarial Training Be Manipulated By Non-Robust Features?

## A   Additional Related Work

In this part, we discuss several independent (or concurrent) works that are closely related to this work.

Zhu et al. [88] study the effect of conventional adversarial training on differentiating noisy labels, while Zhang et al. [85] show that deliberately injected noisy labels may serve as a regularization that alleviates robust overfitting. Our results focus on the clean-label setting and provide evidence that conventional adversarial training can be hindered without modifying the labels.

Yu et al. [80] suggest explaining the success of availability attacks from the perspective of shortcuts. They further adopt pre-trained models to extract useful features for mitigating model reliance on the shortcuts. This direction is orthogonal to ours.

Liu et al. [38] improve the effectiveness of unlearnable examples [28] by generating grayscale perturbations and using data augmentations. They also conclude that conventional adversarial training will prevent a drop in accuracy measured both on clean images and adversarial images. Contrary to them, we show that, both theoretically and empirically, conventional adversarial training can be hindered by hypocritical perturbations, and we further analyze the necessity of enlarging the defense budget to resist stability attacks.

Gao et al. [20] revisit the trade-off between adversarial robustness and backdoor robustness [72]. They conclude that backdoor attacks are ineffective when the defense budget of adversarial training surpasses the trigger magnitude. In contrast, our results indicate that stability attacks are still harmful to adversarial training when the defense budget is not large enough. In a simple statistical setting, a defense budget $\epsilon + \eta$ is necessary (where $\eta$ is a positive number). In the general case, a defense budget of $2\epsilon$ is sufficient. In our experiments, a defense budget of about $1.5\epsilon \sim 1.75\epsilon$ provides the best empirical $\epsilon$-robustness.

Wang et al. [71] argue that it is necessary to use robust features for compromising adversarial training. To this end, they adopt a relatively large attack budget $\epsilon_a = 32/255$ for crafting their poisons (they use one type of adversarial perturbations), and show that their poisons can decrease the performance of the models trained using smaller defense budgets (such as $\epsilon_d = 8/255$ and $\epsilon_d = 16/255$). In contrast, we focus on a more realistic setting that does not require a larger attack budget. We demonstrate that it is possible to hinder adversarial training when $\epsilon_a = \epsilon_d$. Furthermore, we provide both theoretical and empirical results showing how to adapt the defense to maintain robustness.

Fu et al. [19] explore how to protect data privacy against adversarial training. The main purpose of their poisons is to compromise adversarial training by requiring the perturbation budget of their poisons to be larger than that of adversarial training. In this way, they show that the natural accuracy of the adversarially trained models can be largely decreased, let alone robust accuracy. From this perspective, our work is complementary to theirs. We pursue to not increase the attack budget of stability attacks, keeping it as small as possible. We successfully demonstrate that stability attacks are still harmful to conventional adversarial training without enlarging the attack budget. This makes the threat of stability attacks more insidious than that of Fu et al. [19].

On the other hand, we find that our implementation of stability attacks using hypocritical perturbations has some similarities to the robust unlearnable examples in Fu et al. [19]. Specifically, although the robust unlearnable examples are generated via a complicated min-min-max optimization process [19], we notice that their noise generator can be viewed as an adversarially trained model. This implies that the robust error-minimizing (REM) noise [19] might be useful in demonstrating the feasibility of stability attacks. To verify this, we run the source code from the authors with default hyperparameters, and compare our crafted hypocritical perturbation with their generated noise under the setting of stability attacks. For a fair comparison, here we apply a very simple trick called EOT [2] in our method, since the trick is also used by REM [19]. The additional time cost of the EOT trick is very small and negligible.

Our experimental results, shown in Table 8, demonstrate that the robust error-minimizing noise is also effective as stability attacks, though it was originally proposed as a delusive attack. It is noteworthy that the robust accuracy is not evaluated in [19]. In this sense, the effectiveness of REM as an stability attack can be regarded as one of our novel findings. Importantly, our method outperforms REM in terms of the robust accuracy against AutoAttack. Since AutoAttack is the most reliable evaluation metric of model robustness among the test-time attacks [11], this indicates that our method is reliably more effective than REM in degrading model robustness. It is also noteworthy that our method is significantly more efficient than REM, as shown in Table 9. We note that the efficiency of our method is largely due to the fact that our crafting model is fast to train. Specifically, the time cost of training our crafting model is only 0.3 hours, while it takes 20.8 hours for REM. That is, our crafting model is nearly 70 times faster to train than that of REM. In short, our method is not only more effective, but also more efficient, than REM as a stability attack.

Table 8: Comparison with REM [19]: Test robustness (%) of PGD-AT using a defense budget $\epsilon_d = 8/255$ on CIFAR-10. We report mean and standard deviation over 3 random runs.

| Attack | Natural | FGSM | PGD-20 | PGD-100 | CW$_\infty$ | AutoAttack |
|---|---|---|---|---|---|---|
| None (clean) | 82.17 ± 0.71 | 56.63 ± 0.54 | 50.63 ± 0.56 | 50.35 ± 0.59 | 49.37 ± 0.57 | 46.99 ± 0.62 |
| DeepConfuse [16] | 81.25 ± 1.52 | 54.14 ± 0.63 | 48.25 ± 0.40 | 48.02 ± 0.40 | 47.34 ± 0.05 | 44.79 ± 0.36 |
| Unlearnable Examples [28] | 83.67 ± 0.86 | 57.51 ± 0.31 | 50.74 ± 0.37 | 50.31 ± 0.38 | 49.81 ± 0.24 | 47.25 ± 0.32 |
| NTGA [81] | 82.99 ± 0.40 | 55.71 ± 0.36 | 49.17 ± 0.27 | 48.82 ± 0.30 | 47.96 ± 0.16 | 45.36 ± 0.32 |
| Adversarial Poisoning [18] | **77.35 ± 0.43** | 53.93 ± 0.02 | 49.95 ± 0.11 | 49.76 ± 0.08 | 48.35 ± 0.04 | 46.13 ± 0.18 |
| REM [19] | 85.63 ± 1.05 | **42.86 ± 1.09** | 35.40 ± 0.04 | 35.11 ± 0.09 | **35.24 ± 0.33** | 33.09 ± 0.24 |
| Hypocritical Perturbation (ours) | 87.60 ± 0.45 | 45.00 ± 0.77 | **34.89 ± 0.36** | 34.27 ± 0.36 | 36.28 ± 0.38 | **32.79 ± 0.37** |

Table 9: Comparison with REM [19]: Time cost (min) of poisoning CIFAR-10.

| Method | Training the crafting model | Perturbation generation | Total |
|---|---|---|---|
| REM [19] | 1252.4 | 98.1 | 1350.5 |
| Hypocritical Perturbation (ours) | **18.5** | **17.3** | **35.8** |

# B  Omitted Tables

Table 10: Full table of Table 2: Test robustness (%) of PGD-AT using a defense budget $\epsilon_d = 8/255$ on CIFAR-10. We report mean and standard deviation over 3 random runs.

| Attack | Natural | FGSM | PGD-20 | PGD-100 | CW$_\infty$ | AutoAttack |
|---|---|---|---|---|---|---|
| None (clean) | 82.17 ± 0.71 | 56.63 ± 0.54 | 50.63 ± 0.56 | 50.35 ± 0.59 | 49.37 ± 0.57 | 46.99 ± 0.62 |
| DeepConfuse [16] | 81.25 ± 1.52 | 54.14 ± 0.63 | 48.25 ± 0.40 | 48.02 ± 0.40 | 47.34 ± 0.05 | 44.79 ± 0.36 |
| Unlearnable Examples [28] | 83.67 ± 0.86 | 57.51 ± 0.31 | 50.74 ± 0.37 | 50.31 ± 0.38 | 49.81 ± 0.24 | 47.25 ± 0.32 |
| NTGA [81] | 82.99 ± 0.40 | 55.71 ± 0.36 | 49.17 ± 0.27 | 48.82 ± 0.30 | 47.96 ± 0.16 | 45.36 ± 0.32 |
| Adversarial Poisoning [18] | **77.35 ± 0.43** | 53.93 ± 0.02 | 49.95 ± 0.11 | 49.76 ± 0.08 | 48.35 ± 0.04 | 46.13 ± 0.18 |
| Hypocritical Perturbation (ours) | 88.07 ± 1.10 | **47.93 ± 1.88** | **37.61 ± 0.77** | **36.96 ± 0.61** | **38.58 ± 1.15** | **35.44 ± 0.77** |

Table 11: Full table of Table 3: Test robustness (%) of PGD-AT using a defense budget $\epsilon_d = 8/255$ across different datasets. We report mean and standard deviation over 3 random runs.

| Dataset | Attack | Natural | FGSM | PGD-20 | PGD-100 | CW$_\infty$ | AutoAttack |
|---|---|---|---|---|---|---|---|
| SVHN | None | 93.95 ± 0.21 | 71.83 ± 1.10 | 57.15 ± 0.31 | 56.02 ± 0.33 | 54.93 ± 0.19 | 50.50 ± 0.44 |
| | Adv. | **87.50 ± 0.30** | **56.12 ± 0.33** | 46.71 ± 0.25 | 46.32 ± 0.26 | 45.70 ± 0.27 | 42.48 ± 0.21 |
| | Hyp. | 96.06 ± 0.01 | 59.41 ± 0.07 | **38.17 ± 0.19** | **37.29 ± 0.21** | **40.54 ± 0.27** | **35.43 ± 0.29** |
| CIFAR-100 | None | 56.15 ± 0.17 | 31.50 ± 0.16 | 28.38 ± 0.39 | 28.28 ± 0.40 | 26.53 ± 0.27 | 24.30 ± 0.31 |
| | Adv. | **52.14 ± 0.34** | 28.59 ± 0.12 | 26.19 ± 0.11 | 26.09 ± 0.12 | 24.36 ± 0.09 | 22.71 ± 0.11 |
| | Hyp. | 62.22 ± 0.11 | **26.38 ± 0.11** | **21.51 ± 0.06** | **21.13 ± 0.02** | **21.13 ± 0.23** | **18.74 ± 0.10** |
| Tiny-ImageNet | None | **49.34 ± 2.61** | 25.67 ± 0.92 | 22.99 ± 0.37 | 22.86 ± 0.36 | 20.67 ± 0.69 | 18.54 ± 0.61 |
| | Adv. | 49.52 ± 0.19 | 22.93 ± 0.38 | 20.01 ± 0.24 | 19.91 ± 0.24 | 18.75 ± 0.19 | 16.83 ± 0.25 |
| | Hyp. | 55.92 ± 1.95 | **20.21 ± 0.84** | **15.61 ± 0.31** | **15.26 ± 0.26** | **14.99 ± 0.73** | **12.53 ± 0.57** |

Table 12: Full table of Table 5: Test robustness (%) of various adaptive defenses on the hypocritically perturbed CIFAR-10. We report mean and standard deviation over 3 random runs.

| Defense | Natural | FGSM | PGD-20 | PGD-100 | CW$_\infty$ | AutoAttack |
|---|---|---|---|---|---|---|
| PGD-AT ($\epsilon_d = 8/255$) | 88.07 ± 1.10 | 47.93 ± 1.88 | 37.61 ± 0.77 | 36.96 ± 0.61 | 38.58 ± 1.15 | 35.44 ± 0.77 |
| + Random Noise | 87.62 ± 0.07 | 47.46 ± 0.08 | 38.35 ± 0.08 | 37.90 ± 0.07 | 39.07 ± 0.20 | 36.25 ± 0.14 |
| + Gaussian Smoothing | 83.95 ± 0.27 | 50.96 ± 0.24 | 42.80 ± 0.40 | 42.34 ± 0.38 | 42.41 ± 0.19 | 40.07 ± 0.29 |
| + Cutout | **88.26 ± 0.15** | 49.23 ± 0.42 | 39.77 ± 0.26 | 39.25 ± 0.25 | 40.38 ± 0.25 | 37.61 ± 0.35 |
| + AutoAugment | 86.24 ± 1.14 | 48.87 ± 1.01 | 40.19 ± 0.67 | 39.65 ± 0.72 | 37.66 ± 0.88 | 35.07 ± 0.88 |
| PGD-AT ($\epsilon_d = 14/255$) | 80.00 ± 1.91 | 56.86 ± 1.42 | 52.92 ± 0.86 | 52.83 ± 0.86 | **50.36 ± 1.11** | **48.63 ± 0.93** |
| TRADES ($\epsilon_d = 12/255$) | 79.63 ± 0.06 | 55.73 ± 0.04 | 51.77 ± 0.15 | 51.63 ± 0.15 | 48.68 ± 0.06 | 47.83 ± 0.02 |
| MART ($\epsilon_d = 14/255$) | 77.29 ± 0.87 | **57.10 ± 0.57** | **53.82 ± 0.36** | **53.71 ± 0.34** | 49.03 ± 0.47 | 47.67 ± 0.51 |

## C  Proofs

In this section, we provide the proofs of our theoretical results in Section 3 and Section 4.

### C.1  Proof of Proposition 1

**Proposition 1 (restated).** *Let $\epsilon = 2\eta$ and denote by $\mathcal{A}_{\mathrm{adv}}(f)$ the adversarial accuracy, i.e., the probability of a classifier correctly predicting $y$ on the data (7) under $\ell_\infty$ perturbations. Then, we have*

$$\mathcal{A}_{\mathrm{adv}}(f_{\mathrm{nat}}) \leq \Pr\left\{\mathcal{N}(0,1) < \frac{1 - d\eta^2}{\sigma\sqrt{1 + d\eta^2}}\right\}, \quad \mathcal{A}_{\mathrm{adv}}(f_{\mathrm{rob}}) = \Pr\left\{\mathcal{N}(0,1) < \frac{1 - 2\eta}{\sigma}\right\}.$$

*Proof.* Recalling that in Equation (8), we have the natural classifier:

$$f_{\mathrm{nat}}(\boldsymbol{x}) := \mathrm{sign}(\boldsymbol{w}_{\mathrm{nat}}^\top \boldsymbol{x}), \text{ where } \boldsymbol{w}_{\mathrm{nat}} := [1, \eta, \dots, \eta], \tag{12}$$

and in Equation (9), the robust classifier is defined as:

$$f_{\mathrm{rob}}(\boldsymbol{x}) := \mathrm{sign}(\boldsymbol{w}_{\mathrm{rob}}^\top \boldsymbol{x}), \text{ where } \boldsymbol{w}_{\mathrm{rob}} := [1, 0, \dots, 0]. \tag{13}$$

Then, the adversarial accuracy of the natural classifier on the data $\mathcal{D}$ (7) is

$$\begin{aligned}
\mathcal{A}_{\mathrm{adv}}(f_{\mathrm{nat}}) &= 1 - \Pr_{(\boldsymbol{x},y)\sim\mathcal{D}}\left\{\exists\|\boldsymbol{\delta}\|_\infty \leq \epsilon, f_{\mathrm{nat}}(\boldsymbol{x}+\boldsymbol{\delta}) \neq y\right\} \\
&= 1 - \Pr_{(\boldsymbol{x},y)\sim\mathcal{D}}\left\{\min_{\|\boldsymbol{\delta}\|_\infty \leq \epsilon}[y \cdot f_{\mathrm{nat}}(\boldsymbol{x}+\boldsymbol{\delta})] < 0\right\} \\
&= 1 - \Pr\left\{\min_{\|\boldsymbol{\delta}\|_\infty \leq \epsilon}\left[y \cdot \left(1 \cdot \left(\mathcal{N}(y,\sigma^2)+\delta_1\right) + \sum_{i=2}^{d+1}\eta \cdot \left(\mathcal{N}(y\eta,\sigma^2)+\delta_i\right)\right)\right] < 0\right\} \\
&\leq 1 - \Pr\left\{y \cdot \left(1 \cdot \left(\mathcal{N}(y,\sigma^2)\right) + \sum_{i=2}^{d+1}\eta \cdot \left(\mathcal{N}(y\eta,\sigma^2)-\epsilon\right)\right) < 0\right\} \\
&= 1 - \Pr\left\{\mathcal{N}(1,\sigma^2) + \eta\sum_{i=2}^{d+1}\mathcal{N}(\eta-\epsilon,\sigma^2) < 0\right\} \\
&= \Pr\left\{\mathcal{N}(1,\sigma^2) + \eta\sum_{i=2}^{d+1}\mathcal{N}(\eta-\epsilon,\sigma^2) > 0\right\} \\
&= \Pr\left\{\mathcal{N}(0,1) < \frac{1-d\eta^2}{\sigma\sqrt{1+d\eta^2}}\right\}.
\end{aligned} \tag{14}$$

Similarly, the adversarial accuracy of the robust classifier on the data $\mathcal{D}$ (7) is

$$\begin{aligned}
\mathcal{A}_{\mathrm{adv}}(f_{\mathrm{rob}}) &= 1 - \Pr_{(\boldsymbol{x},y)\sim\mathcal{D}}\left\{\exists\|\boldsymbol{\delta}\|_\infty \leq \epsilon, f_{\mathrm{rob}}(\boldsymbol{x}+\boldsymbol{\delta}) \neq y\right\} \\
&= 1 - \Pr_{(\boldsymbol{x},y)\sim\mathcal{D}}\left\{\min_{\|\boldsymbol{\delta}\|_\infty \leq \epsilon}[y \cdot f_{\mathrm{rob}}(\boldsymbol{x}+\boldsymbol{\delta})] < 0\right\} \\
&= 1 - \Pr\left\{\min_{\|\boldsymbol{\delta}\|_\infty \leq \epsilon}\left[y \cdot \left(1 \cdot \left(\mathcal{N}(y,\sigma^2)+\delta_1\right)\right)\right] < 0\right\} \\
&= 1 - \Pr\left\{\min_{\|\boldsymbol{\delta}\|_\infty \leq \epsilon}\left[\mathcal{N}(1,\sigma^2)+\delta_1\right] < 0\right\} \\
&= 1 - \Pr\left\{\mathcal{N}(1,\sigma^2) - \epsilon < 0\right\} \\
&= \Pr\left\{\mathcal{N}(1-\epsilon,\sigma^2) > 0\right\} \\
&= \Pr\left\{\mathcal{N}(0,1) < \frac{1-2\eta}{\sigma}\right\}.
\end{aligned} \tag{15}$$

$\square$

### C.2  Proof of Theorem 1

The following theorems rely on the analytical solution of optimal linear $\ell_\infty$-robust classifier on mixture Gaussian distributions. Concretely, the optimization problem is to minimize the adversarial risk on a distribution $\widehat{\mathcal{D}}$ with a

defense budget $\hat{\epsilon}$:

$$\min_f \mathcal{R}_{\text{adv}}^{\hat{\epsilon}}(f, \widehat{\mathcal{D}}), \quad \text{where} \quad \mathcal{R}_{\text{adv}}^{\hat{\epsilon}}(f, \widehat{\mathcal{D}}) \coloneqq \mathbb{E}_{(\boldsymbol{x},y) \sim \widehat{\mathcal{D}}} \left[ \max_{\|\boldsymbol{\xi}\|_\infty \leq \hat{\epsilon}} \mathbb{1} \left( \text{sign}(\boldsymbol{w}^\top(\boldsymbol{x} + \boldsymbol{\xi}) + b) \neq y \right) \right], \quad (16)$$

where $f(\boldsymbol{x}) = \text{sign}(\boldsymbol{w}^\top \boldsymbol{x} + b)$, and $\mathbb{1}(\cdot)$ denotes the indicator function.

We note that optimal linear robust classifiers have been obtained for certain data distributions in previous work [67, 29, 14, 30, 78, 64]. Here, our goal is to establish similar optimal linear robust classifiers for the classification tasks in our setting. We only employ linear classifiers, since it is highly nontrivial to consider non-linearity for adversarial training on mixture Gaussian distributions [14].

**Lemma 1.** *Assume that the adversarial perturbation in data $\mathcal{T}_{\text{adv}}$ (10) is moderate such that $\eta/2 \leq \epsilon < 1/2$. Then, minimizing the adversarial risk (16) on the data $\mathcal{T}_{\text{adv}}$ with a defense budget $\epsilon$ can result in a classifier that assigns 0 weight to the features $x_i$ for $i \geq 2$.*

*Proof.* We prove the lemma by contradiction.

The goal is to minimize the adversarial risk on the distribution $\mathcal{T}_{\text{adv}}$, which can be written as follows:

$$
\begin{aligned}
\mathcal{R}_{\text{adv}}^{\epsilon}(f, \mathcal{T}_{\text{adv}}) &= \Pr_{(\boldsymbol{x},y) \sim \mathcal{T}_{\text{adv}}} \left\{ \exists \|\boldsymbol{\delta}\|_\infty \leq \epsilon, f(\boldsymbol{x} + \boldsymbol{\delta}) \neq y \right\} \\
&= \Pr_{(\boldsymbol{x},y) \sim \mathcal{T}_{\text{adv}}} \left\{ \min_{\|\boldsymbol{\delta}\|_\infty \leq \epsilon} [y \cdot f(\boldsymbol{x} + \boldsymbol{\delta})] < 0 \right\} \\
&= \Pr_{(\boldsymbol{x},y) \sim \mathcal{T}_{\text{adv}}} \left\{ \max_{\|\boldsymbol{\delta}\|_\infty \leq \epsilon} [f(\boldsymbol{x} + \boldsymbol{\delta})] > 0 \mid y = -1 \right\} \cdot \Pr_{(\boldsymbol{x},y) \sim \mathcal{T}_{\text{adv}}} \{y = -1\} \\
&\quad + \Pr_{(\boldsymbol{x},y) \sim \mathcal{T}_{\text{adv}}} \left\{ \min_{\|\boldsymbol{\delta}\|_\infty \leq \epsilon} [f(\boldsymbol{x} + \boldsymbol{\delta})] < 0 \mid y = +1 \right\} \cdot \Pr_{(\boldsymbol{x},y) \sim \mathcal{T}_{\text{adv}}} \{y = +1\} \\
&= \underbrace{\Pr \left\{ \max_{\|\boldsymbol{\delta}\|_\infty \leq \epsilon} \left[ w_1(\mathcal{N}(\epsilon - 1, \sigma^2) + \delta_1) + \sum_{i=2}^{d+1} w_i(\mathcal{N}(\epsilon - \eta, \sigma^2) + \delta_i) + b \right] > 0 \right\} \cdot \frac{1}{2}}_{\mathcal{R}_{\text{adv}}^{\epsilon}(f, \mathcal{T}_{\text{adv}}^{(-1)})} \\
&\quad + \underbrace{\Pr \left\{ \min_{\|\boldsymbol{\delta}\|_\infty \leq \epsilon} \left[ w_1(\mathcal{N}(1 - \epsilon, \sigma^2) + \delta_1) + \sum_{i=2}^{d+1} w_i(\mathcal{N}(\eta - \epsilon, \sigma^2) + \delta_i) + b \right] < 0 \right\} \cdot \frac{1}{2}}_{\mathcal{R}_{\text{adv}}^{\epsilon}(f, \mathcal{T}_{\text{adv}}^{(+1)})}
\end{aligned}
$$
$$(17)$$

Consider an optimal solution $\boldsymbol{w}$ in which $w_i > 0$ for some $i \geq 2$. Then, we have

$$\mathcal{R}_{\text{adv}}^{\epsilon}(f, \mathcal{T}_{\text{adv}}^{(-1)}) = \Pr \left\{ \underbrace{\sum_{j \neq i} \max_{\|\delta_j\| \leq \epsilon} \left[ w_j(\mathcal{N}(\epsilon - [\boldsymbol{w}_{\text{nat}}]_j, \sigma^2) + \delta_j) + b \right]}_{\mathbb{A}} + \underbrace{\max_{\|\delta_i\| \leq \epsilon} \left[ w_i(\mathcal{N}(\epsilon - \eta, \sigma^2) + \delta_i) \right]}_{\mathbb{B}} > 0 \right\}, \quad (18)$$

where $\boldsymbol{w}_{\text{nat}} \coloneqq [1, \eta, \dots, \eta]$ as in Equation (8). Since $w_i > 0$, $\mathbb{B}$ is maximized when $\delta_i = \epsilon$. Thus, the contribution of terms depending on $w_i$ to $\mathbb{B}$ is a normally-distributed random variable with mean $2\epsilon - \eta$. Since $2\epsilon - \eta \geq 0$, setting $w_i$ to zero can only decrease the risk. This contradicts the optimality of $\boldsymbol{w}$. Formally,

$$\mathcal{R}_{\text{adv}}^{\epsilon}(f, \mathcal{T}_{\text{adv}}^{(-1)}) = \Pr \left\{ \mathbb{A} + w_i \mathcal{N}(2\epsilon - \eta, \sigma^2) > 0 \right\} > \Pr \left\{ \mathbb{A} > 0 \right\}. \quad (19)$$

We can also assume $w_i < 0$ and similar contradiction holds. Therefore, minimizing the adversarial risk on $\mathcal{T}_{\text{adv}}$ leads to $w_i = 0$ for $i \geq 2$. $\qquad \square$

**Lemma 2.** *Assume that the adversarial perturbation in data $\mathcal{T}_{\text{adv}}$ (10) is moderate such that $\eta/2 \leq \epsilon < 1/2$. Then, minimizing the adversarial risk (16) on the data $\mathcal{T}_{\text{adv}}$ with a defense budget $\epsilon$ results in a classifier that assigns a positive weight to the feature $x_1$.*

*Proof.* We prove the lemma by contradiction.

The goal is to minimize the adversarial risk on the distribution $\mathcal{T}_{\text{adv}}$, which has been written in Equation (17).

Consider an optimal solution $\boldsymbol{w}$ in which $w_1 \le 0$. Then, we have

$$\mathcal{R}^\epsilon_{\text{adv}}(f, \mathcal{T}^{(-1)}_{\text{adv}}) = \Pr \left\{ \underbrace{\sum_{j=2}^{d+1} \max_{\|\delta_j\| \le \epsilon} \left[ w_j(\mathcal{N}(\epsilon - \eta, \sigma^2) + \delta_j) + b \right]}_{\mathbb{C}} + \underbrace{\max_{\|\delta_1\| \le \epsilon} \left[ w_1(\mathcal{N}(\epsilon - 1, \sigma^2) + \delta_1) \right]}_{\mathbb{D}} > 0 \right\}. \tag{20}$$

Since $w_1 \le 0$, $\mathbb{D}$ is maximized when $\delta_1 = -\epsilon$. Thus, the contribution of the term depending on $w_1$ to $\mathbb{D}$ is a normally-distributed random variable with mean $-1$. Since the mean is negative, setting $w_1$ to be positive can decrease the risk. This contradicts the optimality of $\boldsymbol{w}$. Formally,

$$\mathcal{R}^\epsilon_{\text{adv}}(f, \mathcal{T}^{(-1)}_{\text{adv}}) = \Pr \left\{ \mathbb{C} + w_1 \mathcal{N}(-\eta, \sigma^2) > 0 \right\} > \Pr \left\{ \mathbb{C} + p \mathcal{N}(-\eta, \sigma^2) > 0 \right\}, \tag{21}$$

where $p > 0$ is any positive number. Therefore, minimizing the adversarial risk on $\mathcal{T}_{\text{adv}}$ leads to $w_1 > 0$. $\quad\square$

**Theorem 1 (restated).** *Assume that the adversarial perturbation in the training data $\mathcal{T}_{adv}$ (10) is moderate such that $\eta/2 \le \epsilon < 1/2$. Then, the optimal linear $\ell_\infty$-robust classifier obtained by minimizing the adversarial risk on $\mathcal{T}_{adv}$ with a defense budget $\epsilon$ is equivalent to the robust classifier (9).*

*Proof.* By Lemma 1 and Lemma 2, we have $w_1 > 0$ and $w_i = 0$ ($i \ge 2$) for an optimal linear $\ell_\infty$-robust classifier. Then, the adversarial risk on the distribution $\mathcal{T}_{\text{adv}}$ can be simplified by solving the inner maximization problem first. Formally,

$$\begin{aligned}
\mathcal{R}^\epsilon_{\text{adv}}(f, \mathcal{T}_{\text{adv}}) &= \Pr_{(\boldsymbol{x},y) \sim \mathcal{T}_{\text{adv}}} \left\{ \exists \|\boldsymbol{\delta}\|_\infty \le \epsilon, f(\boldsymbol{x} + \boldsymbol{\delta}) \ne y \right\} \\
&= \Pr_{(\boldsymbol{x},y) \sim \mathcal{T}_{\text{adv}}} \left\{ \min_{\|\boldsymbol{\delta}\|_\infty \le \epsilon} [y \cdot f(\boldsymbol{x} + \boldsymbol{\delta})] < 0 \right\} \\
&= \Pr_{(\boldsymbol{x},y) \sim \mathcal{T}_{\text{adv}}} \left\{ \max_{\|\boldsymbol{\delta}\|_\infty \le \epsilon} [f(\boldsymbol{x} + \boldsymbol{\delta})] > 0 \mid y = -1 \right\} \cdot \Pr_{(\boldsymbol{x},y) \sim \mathcal{T}_{\text{adv}}} \{y = -1\} \\
&\quad + \Pr_{(\boldsymbol{x},y) \sim \mathcal{T}_{\text{adv}}} \left\{ \min_{\|\boldsymbol{\delta}\|_\infty \le \epsilon} [f(\boldsymbol{x} + \boldsymbol{\delta})] < 0 \mid y = +1 \right\} \cdot \Pr_{(\boldsymbol{x},y) \sim \mathcal{T}_{\text{adv}}} \{y = +1\} \\
&= \Pr \left\{ \max_{\|\boldsymbol{\delta}\|_\infty \le \epsilon} \left[ w_1(\mathcal{N}(\epsilon - 1, \sigma^2) + \delta_1) + b \right] > 0 \right\} \cdot \frac{1}{2} \\
&\quad + \Pr \left\{ \min_{\|\boldsymbol{\delta}\|_\infty \le \epsilon} \left[ w_1(\mathcal{N}(1 - \epsilon, \sigma^2) + \delta_1) + b \right] < 0 \right\} \cdot \frac{1}{2} \\
&= \Pr \left\{ w_1 \mathcal{N}(2\epsilon - 1, \sigma^2) + b > 0 \right\} \cdot \frac{1}{2} \\
&\quad + \Pr \left\{ w_1 \mathcal{N}(1 - 2\epsilon, \sigma^2) + b < 0 \right\} \cdot \frac{1}{2},
\end{aligned} \tag{22}$$

which is equivalent to the natural risk on a mixture Gaussian distribution $\mathcal{D}_{\text{tmp}} : \boldsymbol{x} \sim \mathcal{N}(y \cdot \boldsymbol{\mu}_{\text{tmp}}, \sigma^2 \boldsymbol{I})$, where $\boldsymbol{\mu}_{\text{tmp}} = (1 - 2\epsilon, 0, \dots, 0)$. We note that the Bayes optimal classifier for $\mathcal{D}_{\text{tmp}}$ is $f_{\text{tmp}}(\boldsymbol{x}) = \text{sign}(\boldsymbol{\mu}_{\text{tmp}}^\top \boldsymbol{x})$. Specifically, the natural risk

$$\begin{aligned}
\mathcal{R}^0_{\text{adv}}(f, \mathcal{D}_{\text{tmp}}) &= \Pr_{(\boldsymbol{x},y) \sim \mathcal{D}_{\text{tmp}}} \{f(\boldsymbol{x}) \ne y\} \\
&= \Pr_{(\boldsymbol{x},y) \sim \mathcal{D}_{\text{tmp}}} \{y \cdot f(\boldsymbol{x}) < 0\} \\
&= \Pr \left\{ w_1 \mathcal{N}(2\epsilon - 1, \sigma^2) + b > 0 \right\} \cdot \frac{1}{2} \\
&\quad + \Pr \left\{ w_1 \mathcal{N}(1 - 2\epsilon, \sigma^2) + b < 0 \right\} \cdot \frac{1}{2},
\end{aligned} \tag{23}$$

which is minimized when $w_1 = 1 - 2\epsilon > 0$ and $b = 0$. That is, minimizing the adversarial risk $\mathcal{R}^\epsilon_{\text{adv}}(f, \mathcal{T}_{\text{adv}})$ can lead to an optimal linear $\ell_\infty$-robust classifier $f_{\text{tmp}}(\boldsymbol{x})$. Meanwhile, $f_{\text{tmp}}(\boldsymbol{x})$ is equivalent to the robust classifier (9). This concludes the proof of the theorem. $\quad\square$

## C.3 Proof of Theorem 2

**Lemma 3.** *Minimizing the adversarial risk (16) on the data $\mathcal{T}_{hyp}$ (11) with a defense budget $\epsilon$ results in a classifier that assigns positive weights to the features $x_i$ for $i \ge 1$.*

*Proof.* We prove the lemma by contradiction.

The goal is to minimize the adversarial risk on the distribution $\mathcal{T}_{\text{hyp}}$, which can be written as follows:

$$
\begin{aligned}
\mathcal{R}^\epsilon_{\text{adv}}(f, \mathcal{T}_{\text{hyp}}) &= \Pr_{(\boldsymbol{x},y)\sim\mathcal{T}_{hyp}} \{\exists \|\boldsymbol{\delta}\|_\infty \leq \epsilon, f(\boldsymbol{x}+\boldsymbol{\delta}) \neq y\} \\
&= \Pr_{(\boldsymbol{x},y)\sim\mathcal{T}_{\text{hyp}}} \left\{ \min_{\|\boldsymbol{\delta}\|_\infty\leq\epsilon} [y \cdot f(\boldsymbol{x}+\boldsymbol{\delta})] < 0 \right\} \\
&= \Pr_{(\boldsymbol{x},y)\sim\mathcal{T}_{\text{hyp}}} \left\{ \max_{\|\boldsymbol{\delta}\|_\infty\leq\epsilon} [f(\boldsymbol{x}+\boldsymbol{\delta})] > 0 \mid y = -1 \right\} \cdot \Pr_{(\boldsymbol{x},y)\sim\mathcal{T}_{\text{hyp}}} \{y = -1\} \\
&\quad + \Pr_{(\boldsymbol{x},y)\sim\mathcal{T}_{\text{hyp}}} \left\{ \min_{\|\boldsymbol{\delta}\|_\infty\leq\epsilon} [f(\boldsymbol{x}+\boldsymbol{\delta})] < 0 \mid y = +1 \right\} \cdot \Pr_{(\boldsymbol{x},y)\sim\mathcal{T}_{\text{hyp}}} \{y = +1\} \\
&= \underbrace{\Pr \left\{ \max_{\|\boldsymbol{\delta}\|_\infty\leq\epsilon} \left[ w_1(\mathcal{N}(-1-\epsilon,\sigma^2)+\delta_1) + \sum_{i=2}^{d+1} w_i(\mathcal{N}(-\eta-\epsilon,\sigma^2)+\delta_i) + b \right] > 0 \right\} \cdot \frac{1}{2}}_{\mathcal{R}^\epsilon_{\text{adv}}(f,\mathcal{T}^{(-1)}_{\text{hyp}})} \\
&\quad + \underbrace{\Pr \left\{ \min_{\|\boldsymbol{\delta}\|_\infty\leq\epsilon} \left[ w_1(\mathcal{N}(1+\epsilon,\sigma^2)+\delta_1) + \sum_{i=2}^{d+1} w_i(\mathcal{N}(\eta+\epsilon,\sigma^2)+\delta_i) + b \right] < 0 \right\} \cdot \frac{1}{2}}_{\mathcal{R}^\epsilon_{\text{adv}}(f,\mathcal{T}^{(+1)}_{\text{hyp}})}
\end{aligned}
\tag{24}
$$

Consider an optimal solution $\boldsymbol{w}$ in which $w_i \leq 0$ for some $i \geq 1$. Then, we have

$$
\mathcal{R}^\epsilon_{\text{adv}}(f, \mathcal{T}^{(-1)}_{\text{hyp}}) = \Pr \left\{ \underbrace{\sum_{j\neq i} \max_{\|\delta_j\|\leq\epsilon} \left[ w_j(\mathcal{N}(-[\boldsymbol{w}_{\text{nat}}]_j - \epsilon, \sigma^2)+\delta_j) + b \right]}_{\mathbb{G}} + \underbrace{\max_{\|\delta_i\|\leq\epsilon} \left[ w_i(\mathcal{N}(-[\boldsymbol{w}_{\text{nat}}]_i - \epsilon, \sigma^2)+\delta_i) \right]}_{\mathbb{H}} > 0 \right\},
\tag{25}
$$

where $\boldsymbol{w}_{\text{nat}} := [1, \eta, \ldots, \eta]$ as in Equation (8). Since $w_i \leq 0$, $\mathbb{H}$ is maximized when $\delta_i = -\epsilon$. Thus, the contribution of terms depending on $w_i$ to $\mathbb{H}$ is a normally-distributed random variable with mean $-[\boldsymbol{w}_{\text{nat}}]_i - 2\epsilon$. Since the mean is negative, setting $w_i$ to be positive can decrease the risk. This contradicts the optimality of $\boldsymbol{w}$. Formally,

$$
\mathcal{R}^\epsilon_{\text{adv}}(f, \mathcal{T}^{(-1)}_{\text{adv}}) = \Pr \left\{ \mathbb{G} + w_i \mathcal{N}(-[\boldsymbol{w}_{\text{nat}}]_i - 2\epsilon, \sigma^2) > 0 \right\} > \Pr \left\{ \mathbb{G} + p\mathcal{N}(-[\boldsymbol{w}_{\text{nat}}]_i - 2\epsilon, \sigma^2) > 0 \right\},
\tag{26}
$$

where $p > 0$ is any positive number. Therefore, minimizing the adversarial risk on $\mathcal{T}_{\text{hyp}}$ leads to $w_i > 0$ for $i \geq 1$. $\qquad\square$

**Theorem 2 (restated).** *The optimal linear $\ell_\infty$-robust classifier obtained by minimizing the adversarial risk on the perturbed data $\mathcal{T}_{hyp}$ (11) with a defense budget $\epsilon$ is equivalent to the natural classifier (8).*

*Proof.* By Lemma 3, we have $w_i > 0$ for $i \geq 1$ for an optimal linear $\ell_\infty$-robust classifier. Then, we have

$$
\begin{aligned}
\mathcal{R}_{\text{adv}}^\epsilon(f, \mathcal{T}_{\text{hyp}}) &= \Pr_{(\boldsymbol{x},y) \sim \mathcal{T}_{\text{hyp}}} \{\exists \|\boldsymbol{\delta}\|_\infty \leq \epsilon, f(\boldsymbol{x}+\boldsymbol{\delta}) \neq y\} \\
&= \Pr_{(\boldsymbol{x},y) \sim \mathcal{T}_{\text{hyp}}} \left\{\min_{\|\boldsymbol{\delta}\|_\infty \leq \epsilon} [y \cdot f(\boldsymbol{x}+\boldsymbol{\delta})] < 0\right\} \\
&= \Pr_{(\boldsymbol{x},y) \sim \mathcal{T}_{\text{hyp}}} \left\{\max_{\|\boldsymbol{\delta}\|_\infty \leq \epsilon} [f(\boldsymbol{x}+\boldsymbol{\delta})] > 0 \mid y = -1\right\} \cdot \Pr_{(\boldsymbol{x},y) \sim \mathcal{T}_{\text{hyp}}} \{y = -1\} \\
&\quad + \Pr_{(\boldsymbol{x},y) \sim \mathcal{T}_{\text{hyp}}} \left\{\min_{\|\boldsymbol{\delta}\|_\infty \leq \epsilon} [f(\boldsymbol{x}+\boldsymbol{\delta})] < 0 \mid y = +1\right\} \cdot \Pr_{(\boldsymbol{x},y) \sim \mathcal{T}_{\text{hyp}}} \{y = +1\} \\
&= \Pr \left\{\max_{\|\boldsymbol{\delta}_1\|_\infty \leq \epsilon} [w_1(\mathcal{N}(-1-\epsilon, \sigma^2) + \delta_1)] + \sum_{i=2}^{d+1} \max_{\|\boldsymbol{\delta}_i\|_\infty \leq \epsilon} [w_i(\mathcal{N}(-\eta-\epsilon) + \delta_i)] + b > 0\right\} \cdot \frac{1}{2} \\
&\quad + \Pr \left\{\min_{\|\boldsymbol{\delta}_1\|_\infty \leq \epsilon} [w_1(\mathcal{N}(1+\epsilon, \sigma^2) + \delta_1)] + \sum_{i=2}^{d+1} \min_{\|\boldsymbol{\delta}_i\|_\infty \leq \epsilon} [w_i(\mathcal{N}(\eta+\epsilon) + \delta_i)] + b < 0\right\} \cdot \frac{1}{2} \\
&= \Pr \left\{w_1\mathcal{N}(-1, \sigma^2) + \sum_{i=2}^{d+1} w_i\mathcal{N}(-\eta, \sigma^2) + b > 0\right\} \cdot \frac{1}{2} \\
&\quad + \Pr \left\{w_1\mathcal{N}(1, \sigma^2) + \sum_{i=2}^{d+1} w_i\mathcal{N}(\eta, \sigma^2) + b < 0\right\} \cdot \frac{1}{2},
\end{aligned}
\tag{27}
$$

which is equivalent to the natural risk on the mixture Gaussian distribution $\mathcal{D} : \boldsymbol{x} \sim \mathcal{N}(y \cdot \boldsymbol{w}_{\text{nat}}, \sigma^2 \boldsymbol{I})$, where $\boldsymbol{w}_{\text{nat}} = (1, \eta, \ldots, \eta)$. We note that the Bayes optimal classifier for $\mathcal{D}$ is $f_{\text{nat}}(\boldsymbol{x}) = \text{sign}(\boldsymbol{w}_{\text{nat}}^\top \boldsymbol{x})$. Specifically, the natural risk

$$
\begin{aligned}
\mathcal{R}_{\text{adv}}^0(f, \mathcal{D}) &= \Pr_{(\boldsymbol{x},y) \sim \mathcal{D}} \{f(\boldsymbol{x}) \neq y\} \\
&= \Pr_{(\boldsymbol{x},y) \sim \mathcal{D}} \{y \cdot f(\boldsymbol{x}) < 0\} \\
&= \Pr \left\{w_1\mathcal{N}(-1, \sigma^2) + \sum_{i=2}^{d+1} w_i\mathcal{N}(-\eta, \sigma^2) + b > 0\right\} \cdot \frac{1}{2} \\
&\quad + \Pr \left\{w_1\mathcal{N}(1, \sigma^2) + \sum_{i=2}^{d+1} w_i\mathcal{N}(\eta, \sigma^2) + b < 0\right\} \cdot \frac{1}{2},
\end{aligned}
\tag{28}
$$

which is minimized when $w_1 = 1$, $w_i = \eta$ for $i \geq 2$, and $b = 0$. That is, minimizing the adversarial risk $\mathcal{R}_{\text{adv}}^\epsilon(f, \mathcal{T}_{\text{hyp}})$ can lead to an optimal linear $\ell_\infty$-robust classifier $f_{\text{nat}}(\boldsymbol{x})$, which is equivalent to the natural classifier (8). This concludes the proof of the theorem.

$\square$

## C.4   Proof of Theorem 3

**Lemma 4.** *Minimizing the adversarial risk (16) on the data $\mathcal{T}_{\text{hyp}}$ (11) with a defense budget $\epsilon + \eta$ can result in a classifier that assigns $0$ weight to the features $x_i$ for $i \geq 2$.*

*Proof.* The goal is to minimize the adversarial risk on the distribution $\mathcal{T}_{\text{hyp}}$, which can be written as follows:

$$\mathcal{R}_{\text{adv}}^{\epsilon+\eta}(f, \mathcal{T}_{\text{hyp}}) = \Pr_{(\boldsymbol{x},y)\sim\mathcal{T}_{\text{hyp}}} \{\exists \|\boldsymbol{\delta}\|_\infty \le \epsilon+\eta, f(\boldsymbol{x}+\boldsymbol{\delta}) \ne y\}$$

$$= \Pr_{(\boldsymbol{x},y)\sim\mathcal{T}_{\text{hyp}}} \left\{ \min_{\|\boldsymbol{\delta}\|_\infty \le \epsilon+\eta} [y \cdot f(\boldsymbol{x}+\boldsymbol{\delta})] < 0 \right\}$$

$$= \Pr_{(\boldsymbol{x},y)\sim\mathcal{T}_{\text{hyp}}} \left\{ \max_{\|\boldsymbol{\delta}\|_\infty \le \epsilon+\eta} [f(\boldsymbol{x}+\boldsymbol{\delta})] > 0 \mid y = -1 \right\} \cdot \Pr_{(\boldsymbol{x},y)\sim\mathcal{T}_{\text{hyp}}} \{y = -1\}$$

$$+ \Pr_{(\boldsymbol{x},y)\sim\mathcal{T}_{\text{hyp}}} \left\{ \min_{\|\boldsymbol{\delta}\|_\infty \le \epsilon+\eta} [f(\boldsymbol{x}+\boldsymbol{\delta})] < 0 \mid y = +1 \right\} \cdot \Pr_{(\boldsymbol{x},y)\sim\mathcal{T}_{\text{hyp}}} \{y = +1\}$$

$$= \underbrace{\Pr \left\{ \max_{\|\boldsymbol{\delta}\|_\infty \le \epsilon+\eta} \left[ w_1(\mathcal{N}(-1-\epsilon, \sigma^2)+\delta_1) + \sum_{i=2}^{d+1} w_i(\mathcal{N}(-\eta-\epsilon, \sigma^2)+\delta_i) + b \right] > 0 \right\} \cdot \frac{1}{2}}_{\mathcal{R}_{\text{adv}}^{\epsilon+\eta}(f, \mathcal{T}_{\text{hyp}}^{(-1)})}$$

$$+ \underbrace{\Pr \left\{ \min_{\|\boldsymbol{\delta}\|_\infty \le \epsilon+\eta} \left[ w_1(\mathcal{N}(1+\epsilon, \sigma^2)+\delta_1) + \sum_{i=2}^{d+1} w_i(\mathcal{N}(\eta+\epsilon, \sigma^2)+\delta_i) + b \right] < 0 \right\} \cdot \frac{1}{2}}_{\mathcal{R}_{\text{adv}}^{\epsilon+\eta}(f, \mathcal{T}_{\text{hyp}}^{(+1)})}$$

$$\text{(29)}$$

Consider an optimal solution $\boldsymbol{w}$ in which $w_i > 0$ for some $i \ge 2$. Then, we have

$$\mathcal{R}_{\text{adv}}^{\epsilon+\eta}(f, \mathcal{T}_{\text{hyp}}^{(-1)}) = \Pr \left\{ \underbrace{\sum_{j\ne i} \max_{\|\delta_j\| \le \epsilon+\eta} \left[ w_j(\mathcal{N}(-[\boldsymbol{w}_{\text{nat}}]_j - \epsilon, \sigma^2)+\delta_j) + b \right]}_{\mathbb{I}} + \underbrace{\max_{\|\delta_i\| \le \epsilon+\eta} \left[ w_i(\mathcal{N}(-\eta-\epsilon, \sigma^2)+\delta_i) \right]}_{\mathbb{J}} > 0 \right\},$$

$$\text{(30)}$$

where $\boldsymbol{w}_{\text{nat}} := [1, \eta, \dots, \eta]$. Since $w_i > 0$, $\mathbb{J}$ is maximized when $\delta_i = \epsilon+\eta$. Thus, the contribution of terms depending on $w_i$ to $\mathbb{J}$ is a normally-distributed random variable with mean 0. Thus, setting $w_i$ to zero will not increase the risk. Formally, we have

$$\mathcal{R}_{\text{adv}}^{\epsilon+\eta}(f, \mathcal{T}_{\text{hyp}}^{(-1)}) = \Pr \left\{ \mathbb{I} + w_i \mathcal{N}(0, \sigma^2) > 0 \right\} \ge \Pr \left\{ \mathbb{I} > 0 \right\}. \tag{31}$$

We can also assume $w_i < 0$ and a similar argument holds. Similar arguments also hold for $\mathcal{R}_{\text{adv}}^{\epsilon+\eta}(f, \mathcal{T}_{\text{hyp}}^{(+1)})$. Therefore, minimizing the adversarial risk on $\mathcal{T}_{\text{hyp}}$ can lead to $w_i = 0$ for $i \ge 2$. $\square$

**Theorem 3 (restated).** *The optimal linear $\ell_\infty$-robust classifier obtained by minimizing the adversarial risk on the perturbed data $\mathcal{T}_{\text{hyp}}$ (11) with a defense budget $\epsilon + \eta$ is equivalent to the robust classifier (9). Moreover, any defense budget lower than $\epsilon + \eta$ will yield classifiers that still rely on all the non-robust features.*

*Proof.* By Lemma 4, we have $w_i = 0$ ($i \ge 2$) for an optimal linear $\ell_\infty$-robust classifier. Also, the robust classifier will assign a positive weight to the first feature. This is similar to the case in Lemma 2 and we omit the proof here. Then, we have

$$\mathcal{R}_{\text{adv}}^{\epsilon+\eta}(f, \mathcal{T}_{\text{hyp}}) = \Pr_{(\boldsymbol{x},y)\sim\mathcal{T}_{\text{hyp}}} \{\exists \|\boldsymbol{\delta}\|_\infty \le \epsilon+\eta, f(\boldsymbol{x}+\boldsymbol{\delta}) \ne y\}$$

$$= \Pr_{(\boldsymbol{x},y)\sim\mathcal{T}_{\text{hyp}}} \left\{ \min_{\|\boldsymbol{\delta}\|_\infty \le \epsilon+\eta} [y \cdot f(\boldsymbol{x}+\boldsymbol{\delta})] < 0 \right\}$$

$$= \Pr_{(\boldsymbol{x},y)\sim\mathcal{T}_{\text{hyp}}} \left\{ \max_{\|\boldsymbol{\delta}\|_\infty \le \epsilon+\eta} [f(\boldsymbol{x}+\boldsymbol{\delta})] > 0 \mid y = -1 \right\} \cdot \Pr_{(\boldsymbol{x},y)\sim\mathcal{T}_{\text{hyp}}} \{y = -1\}$$

$$+ \Pr_{(\boldsymbol{x},y)\sim\mathcal{T}_{\text{hyp}}} \left\{ \min_{\|\boldsymbol{\delta}\|_\infty \le \epsilon+\eta} [f(\boldsymbol{x}+\boldsymbol{\delta})] < 0 \mid y = +1 \right\} \cdot \Pr_{(\boldsymbol{x},y)\sim\mathcal{T}_{\text{hyp}}} \{y = +1\} \tag{32}$$

$$= \Pr \left\{ \max_{\|\boldsymbol{\delta}\|_\infty \le \epsilon+\eta} [w_1(\mathcal{N}(-1-\epsilon, \sigma^2)+\delta_1) + b] > 0 \right\} \cdot \frac{1}{2}$$

$$+ \Pr \left\{ \min_{\|\boldsymbol{\delta}\|_\infty \le \epsilon+\eta} [w_1(\mathcal{N}(1+\epsilon, \sigma^2)+\delta_1) + b] < 0 \right\} \cdot \frac{1}{2}$$

$$= \Pr \left\{ w_1 \mathcal{N}(-1-\eta, \sigma^2) + b > 0 \right\} \cdot \frac{1}{2}$$

$$+ \Pr \left\{ w_1 \mathcal{N}(1-\eta, \sigma^2) + b < 0 \right\} \cdot \frac{1}{2},$$

which is equivalent to the natural risk on a mixture Gaussian distribution $\mathcal{D}_{\text{tmp}}$ : $\boldsymbol{x} \sim \mathcal{N}(y \cdot \boldsymbol{\mu}_{\text{tmp}}, \sigma^2 \boldsymbol{I})$, where $\boldsymbol{\mu}_{\text{tmp}} = (1 - \eta, 0, \ldots, 0)$. We note that the Bayes optimal classifier for $\mathcal{D}_{\text{tmp}}$ is $f_{\text{tmp}}(\boldsymbol{x}) = \text{sign}(\boldsymbol{\mu}_{\text{tmp}}^{\top} \boldsymbol{x})$. Specifically, the natural risk

$$
\begin{aligned}
\mathcal{R}_{\text{adv}}^0(f, \mathcal{D}_{\text{tmp}}) &= \Pr_{(\boldsymbol{x}, y) \sim \mathcal{D}_{\text{tmp}}} \{f(\boldsymbol{x}) \neq y\} \\
&= \Pr_{(\boldsymbol{x}, y) \sim \mathcal{D}_{\text{tmp}}} \{y \cdot f(\boldsymbol{x}) < 0\} \\
&= \Pr \left\{ w_1 \mathcal{N}(-1 - \eta, \sigma^2) + b > 0 \right\} \cdot \frac{1}{2} \\
&\quad + \Pr \left\{ w_1 \mathcal{N}(1 - \eta, \sigma^2) + b < 0 \right\} \cdot \frac{1}{2},
\end{aligned}
\tag{33}
$$

which is minimized when $w_1 = 1 - \eta > 0$ and $b = 0$. That is, minimizing the adversarial risk $\mathcal{R}_{\text{adv}}^{\epsilon+\eta}(f, \mathcal{T}_{\text{hyp}})$ can lead to an optimal linear $\ell_\infty$-robust classifier $f_{\text{tmp}}(\boldsymbol{x})$, which is equivalent to the robust classifier (9).

Moreover, when the defense budget $\epsilon_d$ is less than $\epsilon + \eta$, the condition in Lemma 4 no longer holds. Instead, in this case, the robust classifier will assign positive weights to the features (i.e., $w_i > 0$ for $i \geq 1$). This is similar to the case in Lemma 3, and thus we omit the proof here. Consequently, this yields classifiers that still rely on all the non-robust features.

$\square$

## C.5 Proof of Theorem 4

**Theorem 4 (restated).** *For any data distribution and any adversary with an attack budget $\epsilon$, training models to minimize the adversarial risk with a defense budget $2\epsilon$ on the perturbed data is sufficient to ensure $\epsilon$-robustness.*

*Proof.* For clarity, we rewrite the adversarial risk in (2) with a defense budget $\epsilon$ as follows:

$$
\mathcal{R}_{\text{adv}}^{\epsilon}(f, \mathcal{T}) := \sum_{(\boldsymbol{x}, y) \in \mathcal{T}} \left[ \max_{\|\boldsymbol{\delta}\| \leq \epsilon} \mathcal{L}(f(\boldsymbol{x} + \boldsymbol{\delta}), y) \right],
\tag{34}
$$

where $\mathcal{T} = \{(\boldsymbol{x}_i, y_i)\}_{i=1}^n$ denotes the empirical training data.

Consider any adversary with an attack budget $\epsilon$, who can perturb $\boldsymbol{x}$ to $\boldsymbol{x} + \boldsymbol{p}$ such that $\|\boldsymbol{p}\| \leq \epsilon$. Then, the learner will receive a perturbed version of training data $\mathcal{T}' = \{(\boldsymbol{x}_i + \boldsymbol{p}_i, y_i)\}_{i=1}^n$.

For any perturbed data point $(\boldsymbol{x}_i + \boldsymbol{p}_i, y_i)$, we have

$$
\begin{aligned}
\max_{\|\boldsymbol{\delta}\| \leq 2\epsilon} \mathcal{L}(f(\boldsymbol{x}_i + \boldsymbol{p}_i + \boldsymbol{\delta}), y_i) &= \max_{\|\boldsymbol{\delta}\| \leq \epsilon, \|\boldsymbol{\xi}\| \leq \epsilon} \mathcal{L}(f(\boldsymbol{x}_i + \boldsymbol{p}_i + \boldsymbol{\delta} + \boldsymbol{\xi}), y_i) \\
&\geq \max_{\|\boldsymbol{\delta}\| \leq \epsilon} \mathcal{L}(f(\boldsymbol{x}_i + \boldsymbol{p}_i + \boldsymbol{\delta} - \boldsymbol{p}_i), y_i) \\
&= \max_{\|\boldsymbol{\delta}\| \leq \epsilon} \mathcal{L}(f(\boldsymbol{x}_i + \boldsymbol{\delta}), y_i).
\end{aligned}
\tag{35}
$$

By summarizing the training points, we have

$$
\mathcal{R}_{\text{adv}}^{2\epsilon}(f, \mathcal{T}') \geq \mathcal{R}_{\text{adv}}^{\epsilon}(f, \mathcal{T}).
\tag{36}
$$

That is, the adversarial risk with a defense budget $2\epsilon$ on the perturbed data is an upper bound of the adversarial risk with a defense budget $\epsilon$ on the original data. Therefore, a defense budget $2\epsilon$ is sufficient to ensure the learning of $\epsilon$-robustness. $\square$

## D  Experimental Settings

**Adversary capability.**   We focus on the clean-label setting, where an adversary can only provide correctly labeled but misleading training data. In this setting, the main constraint is to craft perturbations as small as possible [16]. Thus, we consider an $\ell_\infty$ adversary with an *attack budget* $\epsilon_a = 8/255$ by following Huang et al. [28], Yuan and Wu [81], Tao et al. [64], Fowl et al. [18]. We note that this constraint is consistent with common research on test-time adversarial examples [1].

**Crafting details.**   We conduct stability attacks by applying the hypocritical perturbation into the training set. Unless otherwise specified, we craft the perturbations by solving the error-minimizing objective (4) with 100 steps of PGD, where a step size of $0.8/255$ is used by following Fowl et al. [18]. Our crafting model is adversarially trained with a *crafting budget* $\epsilon_c = 0.25\epsilon_a$ for 10 epochs before generating perturbations. That is, setting $\epsilon_c = 2/255$ performs best, as shown in Figure 2(a).

**Training details.**   We evaluate the effectiveness of the hypocritical perturbation on benchmark datasets including CIFAR-10/100 [33], SVHN [43], and Tiny-ImageNet [34]. Unless otherwise specified, we use ResNet-18 [26] as the default architecture for both the crafting model and the learning model. For adversarial training, we mainly follow the settings in previous studies [83, 70, 50]. By convention, the *defense budget* is equal to the attack budget, i.e., $\epsilon_d = 8/255$. The networks are trained for 100 epochs using SGD with momentum 0.9, weight decay $5 \times 10^{-4}$, and an initial learning rate of 0.1 that is divided by 10 at the 75-th and 90-th epoch. Early stopping is done with holding out 1000 examples from the training set. Simple data augmentations such as random crop and horizontal flip are applied. The inner maximization problem during adversarial training is solved by 10-steps PGD (PGD-10) with step size $2/255$.

## E  Feature-level Analysis on CIFAR-10

In Section 3.1, we theoretically showed that the hypocritical perturbation can cause the poisoned model to rely more on non-robust features, thus the natural accuracy of the adversarially trained model is increased while the robust accuracy is decreased. In this part, we aim to provide empirical evidence on the role of non-robust features in the success of our poisoning method on a benchmark dataset. In particular, we will demonstrate that our hypocritical perturbation successfully makes the poisoned model learn more non-robust features.

To show this, by following Section 3.2 of Ilyas et al. [29], we construct a training set where the only features that are useful for classification are the non-robust features (that are extracted from the poisoned model). The standard accuracy of the classifier trained on the constructed dataset can reflect how many non-robust features are learned by the poisoned model (denoted as $f$). To accomplish this, we modify each input-label pair $(\boldsymbol{x}, y)$ as follows. We select a target class $t$ uniformly at random among classes. Then, we add a small adversarial perturbation to $\boldsymbol{x}$ as follows:

$$\boldsymbol{x}_{\text{adv}} = \underset{\|\boldsymbol{x}' - \boldsymbol{x}\| \le \epsilon}{\arg\min}\, \ell(f(\boldsymbol{x}'), t).$$

The resulting input-label pairs $(\boldsymbol{x}_{\text{adv}}, t)$ make up the new training set. Since the resulting inputs $\boldsymbol{x}_{\text{adv}}$ are nearly indistinguishable from the originals $\boldsymbol{x}$, the label $t$ assigned to the modified input is simply incorrect to a human observer. Therefore, only the non-robust features in the training set are predictive, while the non-robust features are extracted from the poisoned model.

We compare the model poisoned by our hypocritical perturbation with the baseline model trained on clean data. These two models correspond to the second row and last row in Table 2, respectively. Using these two models, we construct two datasets in the above-mentioned manner, respectively. Then, two new predictors are trained on the two constructed datasets, respectively, and both predictors are evaluated on clean data. Training parameters follow exactly those adopted by Ilyas et al. [29]. Our numerical results are summarized in Table 13.

Table 13: The predictive ability of the non-robust features learned by the poisoned model.

| Model for constructing the training set | Standard accuracy on the original test set (%) |
| --- | --- |
| The baseline model | 27.46 |
| The poisoned model | **56.77** |

As shown in Table 13, the non-robust features learned by the poisoned model are much more predictive than the baseline. This indicates that the effect of our poisoning method on the non-robust features learned by the poisoned model is validated empirically.

# F   Broader Impact

The attack method in this work might be used by an agent in the real world to damage the robust availability of a machine-learning-based system. We discourage this malicious behavior by presenting the threat model of stability attacks to the community. We further propose an adaptive defense to mitigate this issue. The adaptive defense would help to build a more secure and robust machine learning system in the real world. At the same time, the adaptive defense introduces an additional time cost to search for an appropriate defense budget, which might have a negative impact on carbon emission reduction. Furthermore, society should not be overly optimistic about AI safety, since the current studies mostly focus on perturbations bounded by simple norms (e.g., $\ell_\infty$ norm in this paper). There might exist perturbations beyond the $\ell_p$ ball in the real world, and we are still far from complete model robustness.

# G   On the Trade-off between Accuracy and Robustness

An interesting implication of this work is that the hypocritical perturbation exploits the trade-off between standard generalization and adversarial robustness, a phenomenon that has been widely observed in existing works on adversarial training [67, 83, 14, 40, 59, 79].

Prior work mainly observed that adversarial training improves robust accuracy at the cost of natural accuracy *when the training data is clean*. An explanation for the phenomenon is that there are non-robust features in the original dataset, which are predictive yet brittle [67, 29]. Unlike prior work, the trade-off in this work occurs *when the training data is hypocritically perturbed*. Specifically, we make the following observations:

1. When trained on the hypocritically perturbed data, conventional adversarial training produces models with lower robust accuracy but higher natural accuracy (e.g., see Table 2, Table 3, and Table 4).

2. When trained on the hypocritically perturbed data, adversarial training with adaptive budget can improve robust accuracy while reducing natural accuracy (e.g., see Table 5).

These two observations align well with our theoretical analyses in Section 3 and Section 4, respectively. Concretely, our analyses suggest that the hypocritical perturbation works by reinforcing the non-robust features in the original data, so that the models adversarially trained on the manipulated data still rely on the non-robust features. In this way, the natural accuracy of the models increases because the non-robust features are predictive, while the robust accuracy decreases because the non-robust features are brittle. Furthermore, the effectiveness of the adaptive defense lies in the fact that the reinforced non-robust features can be neutralized by enlarging the defense budget of adversarial training. Thus, the adaptive defense improves robustness at the cost of accuracy.

Meanwhile, we note that it would be unsatisfactory that test robustness is improved at the cost of standard generalization. Several improvements have been proposed to alleviate this issue in the case where the training data is clean, such as RST [49], FAT [84], and SCORE [48]. Incorporating these advances would be helpful in resisting stability attacks, and we leave this as future work.

Finally, we remark that the focus of stability attacks is to degrade test robustness. For this reason, we do not impose additional restrictions on their impact on natural accuracy. Having that said, as a method of stability attacks, the hypocritical perturbation is observed to improve natural accuracy while reducing robust accuracy. We note that this makes stability attacks more insidious. For example, if a poisoned model exhibits higher natural accuracy, practitioners would be more easily enticed to deploy it in a real-world system. However, as its robust accuracy is actually undesirably low, the system is prone to losing its normal function when encountering test-time perturbations. In short, the negative impacts of stability attacks are serious, even with higher natural accuracy. Thus, it is imperative to design better defense methods to mitigate the threat of stability attacks.