# OpenReview forum: "Can Adversarial Training Be Manipulated By Non-Robust Features?"
_NeurIPS.cc/2022/Conference — NeurIPS 2022 Accept_

### Official Review · Reviewer_UnFX · 2022-06-30

**Rating:** 6
**Confidence:** 2
**Soundness:** 3 good
**Presentation:** 4 excellent
**Contribution:** 3 good

**Summary:**

### Summary:

This paper introduces a novel data poisoning attack against adversarial training called *stability attacks*.
The goal is to temper the training data such that the robust performance of adversarial training over this manipulated dataset is degraded.
To construct this attack, a *hypocritical perturbation* is built: unlike *adversarial perturbations*, the aim of *hypocritical perturbations* is to reinforce the non-robust features in the training data.
These perturbations can be generated by negating adversarial example generation objectives.

**Motivation:**
The paper motivates stability attacks from the perspective of robust vs. non-robust features.
Specifically, a simple binary classification task over a mixture of Gaussians is considered.
Statistical analysis on this task shows that adversarial training over *hypocritically perturbed data* is destructive to adversarial robustness.
Moreover, it is shown that a larger perturbation magnitude is needed to guard adversarial training against stability attacks.

**Implementation:**
The effectiveness of stability attacks against adversarial training is demonstrated through extensive experimental results.

**Questions:**

### Questions:
As mentioned above, there are a few questions:

1. How the *hypocritical perturbation* objective is derived? Are there any other objectives that can be used?
2. What is the relationship between this work and prior work on the trade-off between clean and adversarial accuracy?
3. While increasing the perturbation bound would help combat stability attacks, how can one maintain the clean accuracy as the current solution jeopardizes this?

**Limitations:**

A potential discussion on the real-world negative impacts of the current work is missing. This reviewer would encourage the authors to discuss this matter explicitly.

**Strengths And Weaknesses:**

### Strengths:
- The paper is clear, and it guides the reader skillfully.

- The paper is well-motivated.
The statistical analysis of the binary classification task is thorough, and the implications of the theoretical results are discussed comprehensively.

- This paper sheds light on the implications of robust vs. non-robust features from a novel perspective and utilizes these studies to introduce a new threat against adversarial training.

- The experimental settings are discussed in detail, and the effects of different hyper-parameters and architectures on the performance are investigated.

### Weaknesses:
- While the theoretical justifications of *hypocritical perturbations* on the binary classification task are discussed, the relationship of these results with the attack generation process (Eq. (10)) is obscure.
Although the given example for the logistic loss and the binary classification task is appreciated, the origins of the objective function in Eq. (10) need a better justification.

-  Furthermore, a thorough discussion on the relationship of this work with existing works on the trade-off between the clean and robust accuracy of neural networks with adversarial training seems missing.
As the experimental results suggest, the implications align with the observations of Tsipras et al. [63] on the trade-off between the clean and robust accuracy (e.g., see Table 4).
From this perspective, it seems like stability attacks are somehow just exploiting this trade-off to pose their threat on adversarial training.
Thus, a comprehensive discussion on the differences between this work and prior work in this area is required.

---

> ### Author Response · Authors · 2022-08-02
> **Response to Reviewer UnFX (Part 3)**
>
> **References**
>
> [R1] Tao, L., Feng, L., Yi, J., & Chen, S. (2020). With False Friends Like These, Who Can Notice Mistakes?. *arXiv preprint arXiv:2012.14738*.
> [R2] Tao, L., Feng, L., Yi, J., Huang, S. J., & Chen, S. (2021). Better safe than sorry: Preventing delusive adversaries with adversarial training. *Advances in Neural Information Processing Systems*, *34*, 16209-16225.
> [R3] Huang, H., Ma, X., Erfani, S. M., Bailey, J., & Wang, Y. (2020, September). Unlearnable Examples: Making Personal Data Unexploitable. In *International Conference on Learning Representations*.
> [R4] Fu, S., He, F., Liu, Y., Shen, L., & Tao, D. (2021, September). Robust unlearnable examples: Protecting data privacy against adversarial learning. In *International Conference on Learning Representations*.
> [R5] Zhang, J., Xu, X., Han, B., Niu, G., Cui, L., Sugiyama, M., & Kankanhalli, M. (2020, November). Attacks which do not kill training make adversarial learning stronger. In *International conference on machine learning* (pp. 11278-11287). PMLR.
> [R6] Pang, T., Lin, M., Yang, X., Zhu, J., & Yan, S. (2022). Robustness and Accuracy Could Be Reconcilable by (Proper) Definition. *arXiv preprint arXiv:2202.10103*.
> [R7] Raghunathan, A., Xie, S. M., Yang, F., Duchi, J., & Liang, P. (2020, November). Understanding and Mitigating the Tradeoff between Robustness and Accuracy. In *International Conference on Machine Learning* (pp. 7909-7919). PMLR.

---

> > ### Comment · Reviewer_UnFX · 2022-08-05
> > **Post Rebuttal Comments**
> >
> > First, I would like to thank the author(s) for providing a detailed response to all the reviews.
> > The new arrangement of the paper now helps the reader better understand the origins of hypocritical perturbations.
> >
> > I have another follow-up question: reading through the review J69n and your answers, it is mentioned that REM [1] has a similar effect on robust accuracy.
> > To showcase the differences, you mention that the perturbation generation of REM takes around 23 hours, while for yours, it takes around 0.5 hours. This strikes odd to me as an adversarially trained model is used to generate the proposed hypocritical perturbations.
> > Are these reported times only for perturbation generation? Or in total (training the crafting model + perturbation generation)? And is the time calculation consistent for REM [1] and this paper's proposed objective? It would be a great help if you could provide a detailed breakdown of how the time is computed and what it entails.
> >
> > [1] Fu, S., He, F., Liu, Y., Shen, L., & Tao, D. (2021, September). Robust unlearnable examples: Protecting data privacy against adversarial learning. In _ICLR_.

---

> > > ### Author Response · Authors · 2022-08-05
> > > **Response to Post Rebuttal Comments**
> > >
> > > Thank you for your time, suggestions, and for carefully reading and considering our response and others' feedback.
> > >
> > > Following your suggestion, we present a detailed breakdown of the time cost of poisoning. The implementation of an attack consists of two steps: i) pre-training the crafting model, and ii) generating the perturbations using the crafting model. The time cost for each step and the total time cost on CIFAR-10 are reported as follows.
> > >
> > > | Method                           | Training the crafting model (min) | Perturbation generation (min) | Total (min) |
> > > | -------------------------------- | --------------------------------- | ----------------------------- | ----------- |
> > > | REM (Fu et al. [1])              | 1252.4                            | 98.1                          | 1350.5      |
> > > | Hypocritical Perturbation (ours) | **18.5**                          | **17.3**                      | **35.8**    |
> > >
> > > This indicates that the efficiency of our method is largely attributed to the small cost of training the crafting model. Specifically, our crafting model only takes 0.3 hours to train, compared to 20.8 hours for REM (nearly 70 times slower). Notably, the crafting model of REM is trained via a complicated and inefficient three-level optimization process, while ours is obtained with adversarial training. Although adversarial training is known to be more time-consuming than standard training, our crafting model only needs to be adversarially trained for 10 epochs, as our ablation study in Table 2(b) shows that this performs best. This makes our method very efficient.
> > >
> > > We have updated our manuscript to include this detailed comparison in Appendix A.
> > >
> > > In summary, our method is not only more effective, but also more efficient, than REM as a stability attack. More importantly, the main findings and conclusions of this work are fundamentally different from those of Fu et al. [1] because the threat model of stability attacks has never been studied in Fu et al. [1].
> > >
> > > [1] Fu, S., He, F., Liu, Y., Shen, L., & Tao, D. (2021, September). Robust unlearnable examples: Protecting data privacy against adversarial learning. In *ICLR*.

---

> ### Author Response · Authors · 2022-08-02
> **Response to Reviewer UnFX (Part 2)**
>
> > As the experimental results suggest, the implications align with the observations of Tsipras et al. [63] on the trade-off between the clean and robust accuracy (e.g., see Table 4). From this perspective, it seems like stability attacks are somehow just exploiting this trade-off to pose their threat on adversarial training. Thus, a comprehensive discussion on the differences between this work and prior work in this area is required.
> > What is the relationship between this work and prior work on the trade-off between clean and adversarial accuracy?
>
> Thanks for this question. We totally agree that the proposed method for stability attacks exploits the trade-off. Our analysis in Section 3 also implies this phenomenon. Theoretically, there are four cases:
>
> | Case | Training data            | Learning scheme                     | The resulting model |
> | ---- | ------------------------ | ----------------------------------- | ------------------- |
> | 1    | Clean                    | Natural training                    | Natural classifier  |
> | 2    | Clean                    | Adversarial training (conventional) | Robust classifier   |
> | 3    | Hypocritically perturbed | Adversarial training (conventional) | Natural classifier  |
> | 4    | Hypocritically perturbed | Adversarial training (our improved) | Robust classifier   |
>
> Prior work by Tsipras et al. [63] focused on analyzing the trade-off between Case #1 and Case #2, while Case #3 and Case #4 are for the first time depicted by Theorem 2 and Theorem 3 in this work.
>
> - Specifically, Tsipras et al. [63] showed that adversarial training produces the robust classifier by avoiding model reliance on non-robust features (Case #2). Though its robust accuracy is higher, its clean accuracy is lower than that of the natural classifier. This is because the non-robust features are predictive but brittle.
> - In contrast, Case #3 indicates that conventional adversarial training can still rely on non-robust features, if the training data is hypocritically perturbed. In other words, the hypocritical perturbation exactly exploits the trade-off by the means of non-robust features. The hypocritical perturbation can increase model reliance on the non-robust features, which leads to higher clean accuracy, but lower robust accuracy.
>
> **[Update: 2022.08.04]** Following your suggestion, we have added a detailed discussion on the trade-off between the clean and robust accuracy in Appendix G of the updated manuscript.
>
> ---
>
> > While increasing the perturbation bound would help combat stability attacks, how can one maintain the clean accuracy as the current solution jeopardizes this?
>
> Thanks for your question. This is a promising direction to which our proposed defense will be extended. We note that when the training data is clean, there were several improvements in adversarial training to improve clean accuracy while maintaining robust accuracy, such as FAT [R5], SCORE [R6], and RST [R7]. For example, robust self-training (RST) is a variant of adversarial training that can improve robust accuracy without sacrificing clean accuracy by leveraging extra unlabeled data [R7]. In our setting, increasing the defense budget of RST would be helpful in resisting stability attacks. We leave this as future work.
>
> ---
>
> > A potential discussion on the real-world negative impacts of the current work is missing. This reviewer would encourage the authors to discuss this matter explicitly.
>
> Thanks for this suggestion. We have added a section entitled "Broader Impact" to Appendix F of the updated manuscript. The content of Appendix F is as follows:
>
> The attack method in this work might be used by an agent in the real world to damage the robust availability of a machine-learning-based system. We discourage this malicious behavior by presenting the threat model of stability attacks to the community. We further propose an adaptive defense to mitigate this issue. The adaptive defense would help to build a more secure and robust machine learning system in the real world. At the same time, the adaptive defense introduces an additional time cost to search for an appropriate defense budget, which might have a negative impact on carbon emission reduction. Furthermore, society should not be overly optimistic about AI safety, since the current studies mostly focus on perturbations bounded by simple norms (e.g., $\ell_{\infty}$ norm in this paper). There might exist perturbations beyond the $\ell_p$ ball in the real world, and we are still far from complete model robustness.

---

> ### Author Response · Authors · 2022-08-02
> **Response to Reviewer UnFX (Part 1)**
>
> Thank you very much for recognizing our work and the great comments. We answer your questions below, but let us know if further questions remain.
>
> ---
>
> > Although the given example for the logistic loss and the binary classification task is appreciated, the origins of the objective function in Eq. (10) need a better justification.
> > How the *hypocritical perturbation* objective is derived?
>
> This is a very good question. The objective of the hypocritical perturbation is originally proposed in [R1], which aims to correct the error of a pre-trained model $f$ by perturbing the input: $\min_{\|\delta\|\le\epsilon} \mathbb{1}(f(x+\delta)\neq y)$. In [R1], the commonly used cross entropy (CE) loss is leveraged as the surrogate loss for the 0-1 loss: $\min_{\|\delta\|\le\epsilon} \ell(f(x+\delta), y)$.
>
> [R2] applied the hypocritical perturbation proposed by [R1] to the training data, and showed that it can be an effective delusive attack. The hypocritical perturbation is straightforward to craft [R2]:
>
> &emsp;&emsp;&emsp;&emsp;&emsp;&emsp;&emsp;&emsp;$\min_{\|p_i\|\le\epsilon} \ell(f'(x_i+p_i), y_i), \text{ where } f'=\arg\min_{f} \sum_{i=1}^n \ell(f(x_i), y_i).$
>
> [R3] independently designed a similar objective to generate error-minimizing noises (whose goal is the same as delusive attacks, and the difference is that their noise generator $f'$ is trained via a bi-level optimization process):
>
> &emsp;&emsp;&emsp;&emsp;&emsp;&emsp;&emsp;&emsp;$\min_{\|p_i\|\le\epsilon} \ell(f'(x_i+p_i), y_i), \text{ where } f'=\arg\min_{f} \sum_{i=1}^n \min_{\|\delta_i\|\le\epsilon} \ell(f(x_i+\delta_i), y_i).$
>
> This work proposes to adopt an adversarially trained model as the crafting model $f'$. Formally, our poisons are generated as follows:
>
> &emsp;&emsp;&emsp;&emsp;&emsp;&emsp;&emsp;&emsp;$\min_{\|p_i\|\le\epsilon} \ell(f'(x_i+p_i), y_i), \text{ where } f'=\arg\min_{f} \sum_{i=1}^n \max_{\|\delta_i\|\le\epsilon/4} \ell(f(x_i+\delta_i), y_i).$
>
> Following your suggestion, we have added a more detailed description of the hypocritical perturbation,
> the adversarial perturbation, and their origins in Section 2.1 (Preliminaries) of the updated manuscript.
>
> ---
>
> > Are there any other objectives that can be used?
>
> Yes. Table 7 in Appendix A shows that a more complicated objective proposed by [R4] can also be used as an effective stability attack. The method called REM [R4] generates poisons as follows:
>
> &emsp;&emsp;&emsp;&emsp;$\min_{\|p_i\|\le\epsilon} \max_{\|\delta_i\|\le\rho}\ell(f'(x_i+p_i+\delta_i),y_i), \text{ where } f' = \arg\min_{f} \sum_{i=1}^n \min_{\|p_i\|\le\epsilon} \max_{\|\delta_i\|\le\rho} \ell(f(x_i+p_i+\delta_i), y_i).$
>
> It is worth noting that REM was originally proposed as a method for delusive attacks. Our contribution here is to experimentally show that REM is also effective as a method for stability attacks.
>
> Although REM is comparable in effectiveness to our method under the setting of stability attacks, it is computationally expensive. For example, the time cost of REM to poison CIFAR-10 is about 23 hours, while our method requires only 0.5 hours. Put simply, our method is significantly more efficient than REM.

---

### Official Review · Reviewer_J69n · 2022-07-10

**Rating:** 5
**Confidence:** 4
**Soundness:** 2 fair
**Presentation:** 3 good
**Contribution:** 2 fair

**Summary:**

This paper introduces the problem of adversarial training when they face the new type of attacks called stability attacks. The stability attacks aim to compromise the robust availability by slightly manipulating the training data. Most of existing methods neglect that the test robustness of the adversarial trained models when they are under the training-time availability attacks. Under the threat of the stability attacks, they demonstrate that the adversarial trained network with epsilon perturbation budget is not enough to defend against the epsilon bounded adversarial perturbation. The authors argue that it is necessary to enlarge the epsilon perturbation budget when they conduct the adversarial training.

**Questions:**

According to the paper, stability attack consists of training-time perturbation and test-time perturbation in Table 1. In Table 2, the authors compare the stability attacks with other training-time availability attacks (DeepConfuse, Unlearnable Examples, NTGA …). However, the training-time availability attacks have training-time perturbation only. Therefore, the test-time perturbation of FGSM, PGD-20, PGD-100, CW, and AutoAttack in Table 2 should be different for the stability attacks and for other training-time availability attacks. What is the exact experiment setting for the test-time perturbation under the threat of stability attacks?

**Limitations:**

Yes, the authors have addressed the limitations and potential negative societal impact of their work.

**Strengths And Weaknesses:**

Strengths :
1. Clear writing. Easy to understand and well-organized paper.
2. Important experimental result. The test-robustness of adversarial trained network against evasion attacks when they are under the delusive attacks is intriguing result in the adversarial research.

Weaknesses :
1. Low originality: Missing comparison with the recent key reference [Ref_1] which is somewhat similar method of the training-time availability attacks. The attack generation algorithm of training-time and test-time perturbation is similar to [Ref_1]. Furthermore, [Ref_1] achieved the state-of-the-art attack performance against adversarial trained network on clean data. Thus, it is unclear that this paper made the fair comparison between current SOTA poisoning attacks on adversarial trained network. In this regard, the novelty of the threat model on the adversarial trained network by perturbing the training data is marginal.
2. Insufficient explanations on the relationship with non-robust features. This paper passes the buck to the non-robust feature for the experiment results of the increase of standard accuracy and the decrease of robust accuracy when they are under the stability attacks. The authors emphasize the responsibility of the non-robust feature in the title, abstract, and throughout the paper. However, I am not convinced that non-robust feature is the only reason for the experiment result in Table 2. Natural test robustness has increased and test robustness under evasion attacks has decreased when they are adversarially trained under the stability attacks. But, considering the fact that there always exists a trade-off between the standard accuracy and the robust accuracy [Ref_2], non-robust feature can’t be solely blamed. As the authors followed the process of theoretical analysis with the [Ref_3], they need to present another empirical evidence, feature-level analysis, or visualizations for the explanations on relationship with non-robust features as [Ref_3] did.
3. Confounding usage of the term for the similar concept. It is very confused when the term ‘Hyp’, ‘stability attacks’, and similar concept appears throughout the paper.

[Ref_1] Fu, S., He, F., Liu, Y., Shen, L., & Tao, D. (2021, September). Robust unlearnable examples: Protecting data privacy against adversarial learning. In International Conference on Learning Representations.

[Ref_2] Zhang, H., Yu, Y., Jiao, J., Xing, E., El Ghaoui, L., & Jordan, M. (2019, May). Theoretically principled trade-off between robustness and accuracy. In International conference on machine learning (pp. 7472-7482). PMLR.

[Ref_3] Tsipras, D., Santurkar, S., Engstrom, L., Turner, A., & Madry, A. (2018). Robustness may be at odds with accuracy. arXiv preprint arXiv:1805.12152.

---

> ### Author Response · Authors · 2022-08-02
> **Response to Reviewer J69n (Part 4)**
>
> > In Table 2, the authors compare the stability attacks with other training-time availability attacks (DeepConfuse, Unlearnable Examples, NTGA …). However, the training-time availability attacks have training-time perturbation only. Therefore, the test-time perturbation of FGSM, PGD-20, PGD-100, CW, and AutoAttack in Table 2 should be different for the stability attacks and for other training-time availability attacks. What is the exact experiment setting for the test-time perturbation under the threat of stability attacks?
>
> You are right about delusive attacks: they have training-time perturbations only. Thus, delusive attacks are not capable of applying test-time perturbations. Although the poisoning methods we compare in Table 2 were originally proposed for delusive attacks, this work treats them as candidate methods of stability attacks for fair comparison.
>
> To this end, the effectiveness of previous methods in degrading robust accuracy should be evaluated. Since the robust accuracy is NP-hard to compute [Ref_9], it is common practice to use various test-time attacks as a proxy for evaluating test robustness. Therefore, various test-time attacks including FGSM, PGD-20, PGD-100, CW, and AutoAttack are applied for evaluating all methods.
>
> We sincerely thank you for pointing this out. With your comments in mind, we have rechecked the experiment section, and replaced several expressions "stability attacks" with "hypocritical perturbation" in the updated manuscript to avoid ambiguity.
>
> ---
>
> **References**
>
> [Ref_1] Fu, S., He, F., Liu, Y., Shen, L., & Tao, D. (2021, September). Robust unlearnable examples: Protecting data privacy against adversarial learning. In International Conference on Learning Representations.
> [Ref_2] Zhang, H., Yu, Y., Jiao, J., Xing, E., El Ghaoui, L., & Jordan, M. (2019, May). Theoretically principled trade-off between robustness and accuracy. In International conference on machine learning (pp. 7472-7482). PMLR.
> [Ref_3] Tsipras, D., Santurkar, S., Engstrom, L., Turner, A., & Madry, A. (2018, September). Robustness May Be at Odds with Accuracy. In International Conference on Learning Representations.
> [Ref_4] Madry, A., Makelov, A., Schmidt, L., Tsipras, D., & Vladu, A. (2018, February). Towards Deep Learning Models Resistant to Adversarial Attacks. In International Conference on Learning Representations.
> [Ref_5] Tao, L., Feng, L., Yi, J., Huang, S. J., & Chen, S. (2021). Better safe than sorry: Preventing delusive adversaries with adversarial training. Advances in Neural Information Processing Systems, 34, 16209-16225.
> [Ref_6] Fowl, L., Goldblum, M., Chiang, P. Y., Geiping, J., Czaja, W., & Goldstein, T. (2021). Adversarial Examples Make Strong Poisons. Advances in Neural Information Processing Systems, *34*, 30339-30351.
> [Ref_7] Yu, D., Zhang, H., Chen, W., Yin, J., & Liu, T. Y. (2021). Indiscriminate poisoning attacks are shortcuts. arXiv preprint arXiv:2111.00898.
> [Ref_8] He, H., Zha, K., & Katabi, D. (2022). Indiscriminate Poisoning Attacks on Unsupervised Contrastive Learning. arXiv preprint arXiv:2202.11202.
> [Ref_9] Katz, G., Barrett, C., Dill, D. L., Julian, K., & Kochenderfer, M. J. (2017, July). Reluplex: An efficient SMT solver for verifying deep neural networks. In International conference on computer aided verification (pp. 97-117). Springer, Cham.
> [Ref_10] Ilyas, A., Santurkar, S., Tsipras, D., Engstrom, L., Tran, B., & Madry, A. (2019). Adversarial examples are not bugs, they are features. Advances in neural information processing systems, 32.

---

> > ### Comment · Reviewer_J69n · 2022-08-09
> > **Thank authors for the reply**
> >
> > Thank the author for the detailed explanation. The additional results and explanations in the updated Appendix E and G address my initial concern. Thank you for clarifying the contributions of the proposed method of the test-time robustness compared to [Ref_1]. Based on the new results with a fair comparison with [Ref_1] that demonstrates the proposed method outperforms [Ref_1] in both test-time robustness and efficiency, I would like to increase my score from 4 to 5. I expect the authors to include the results in the revised version.
> >
> > [Ref_1] Fu, S., He, F., Liu, Y., Shen, L., & Tao, D. (2021, September). Robust unlearnable examples: Protecting data privacy against adversarial learning. In International Conference on Learning Representations.

---

> > > ### Author Response · Authors · 2022-08-09
> > > **Thanks for your response**
> > >
> > > Dear Reviewer J69n,
> > >
> > > We are glad to hear that we addressed your concerns. Thank you for all your insightful comments and suggestions. We will incorporate your suggestions in the revision.
> > >
> > > Sincerely,
> > > Paper11926 Authors

---

> ### Author Response · Authors · 2022-08-02
> **Response to Reviewer J69n (Part 3)**
>
> > I am not convinced that non-robust feature is the only reason for the experiment result in Table 2. Natural test robustness has increased and test robustness under evasion attacks has decreased when they are adversarially trained under the stability attacks. But, considering the fact that there always exists a trade-off between the standard accuracy and the robust accuracy [Ref_2], non-robust feature can’t be solely blamed.
>
> Thanks for this question. We would like to point out that the trade-off itself can be naturally attributed to the presence of non-robust features in the original dataset [Ref_3] [Ref_10], and our hypocritical perturbation actually exploits the trade-off. The logic is that by applying hypocritical perturbations, non-robust features in the training data are reinforced. Then, the poisoned model will rely more on non-robust features. Thus, its robust accuracy will be lower. Meanwhile, non-robust features are known to be helpful to standard accuracy [Ref_3] [Ref_10], so it is reasonable that the standard accuracy will be higher.
>
> **[Update: 2022.08.04]** To explain the connection between the trade-off and the non-robust features, we have added this discussion in Appendix G of the updated manuscript.
>
> ---
>
> > As the authors followed the process of theoretical analysis with the [Ref_3], they need to present another empirical evidence, feature-level analysis, or visualizations for the explanations on relationship with non-robust features as [Ref_3] did.
>
> Thanks for the suggestion. The authors of [Ref_3] gave a feature-level analysis on CIFAR-10 in [Ref_10]. We thus follow their analysis to provide evidence on the role of non-robust features in the success of our poisoning method. In particular, we demonstrate that our hypocritical perturbation successfully makes the poisoned model learn more non-robust features. We have added a section entitled "Feature-level Analysis on CIFAR-10" to Appendix E of the updated manuscript. The content of Appendix E is as follows:
>
> By following Section 3.2 in [Ref_10], we construct a training set where the only features that are useful for classification are the non-robust features (that are extracted from the poisoned model). The standard accuracy of the classifier trained on the constructed dataset can reflect how many non-robust features are learned by the poisoned model (denoted as $f$). To accomplish this, we modify each input-label pair $(x, y)$ as follows. We select a target class $t$ uniformly at random among classes. Then, we add a small adversarial perturbation to $x$ as follows: $x_{adv} = \arg\min_{\|x'-x\|\le\epsilon} \ell(f(x'), t)$. The resulting input-label pairs $(x_{adv}, t)$ make up the new training set. Since the resulting inputs $x_{adv}$ are nearly indistinguishable from the originals $x$, the label $t$ assigned to the modified input is simply incorrect to a human observer. Therefore, only the non-robust features in the training set are predictive, while the non-robust features are extracted from the poisoned model.
>
> We compare the model poisoned by our hypocritical perturbation with the baseline model trained on clean data. These two models correspond to the second row and last row in Table 2 of the submission, respectively. Using these two models, we construct two datasets in the above-mentioned manner, respectively. Then, two new predictors are trained on the two constructed datasets, respectively, and both predictors are evaluated on clean data. Results are summarized as follows:
>
> | Constructed training set            | Standard accuracy on the original test set (%) |
> | ----------------------------------- | ---------------------------------------------- |
> | Constructed with the baseline model | 27.46                                          |
> | Constructed with the poisoned model | **56.77**                                      |
>
> As shown in the table, the non-robust features learned by the poisoned model are much more predictive than the baseline. This indicates that the effect of our poisoning method on the non-robust features learned by the poisoned model is validated empirically.
>
> ---
>
> > Confounding usage of the term for the similar concept. It is very confused when the term ‘Hyp’, ‘stability attacks’, and similar concept appears throughout the paper.
>
> Sorry for the confusion. We would like to clarify that 'Hyp' denotes 'hypocritical perturbation', and 'stability attack' is a general term for a class of attacks. 'Hypocritical perturbation' is a method that belongs to stability attacks.
>
> To avoid ambiguity, we have made the following changes in the updated manuscript.:
>
> - In Table 2: "Stability Attacks" -> "Hypocritical Perturbation"
> - In Table 8: "Stability Attacks" -> "Hypocritical Perturbation"
> - In Table 11: "Hypocritical Features" -> "Hypocritical Perturbation"

---

> ### Author Response · Authors · 2022-08-02
> **Response to Reviewer J69n (Part 2)**
>
> > [Ref_1] achieved the state-of-the-art attack performance against adversarial trained network on clean data.
>
> Thanks for this question. We would like to point out that this work mainly concerns test robustness. The attack performance on clean data is not the focus of stability attacks in this work.
>
> Having that said, it is still worth noting that *only* when $\rho < \epsilon$, [Ref_1] can achieve SOTA attack performance on clean data. However, as mentioned above, a more reasonable and popular setting is $\rho=\epsilon$ [Ref_5] [Ref_6] [Ref_7] [Ref_8]. In this setting, [Ref_1] will fail to achieve SOTA.
>
> For example, on CIFAR-10, when both $\rho$ and $\epsilon$ are set to $8/255$, the method proposed by [Ref_1] performs poorly in degrading the standard accuracy. The experimental results are reported below:
>
> | Delusive attacks (aimed at degrading standard accuracy) | Standard accuracy (%) |
> | ------------------------------------------------------- | --------------------- |
> | None (clean)                                            | 82.17 ± 0.71          |
> | DeepConfuse                                             | 81.25 ± 1.52          |
> | Unlearnable Examples                                    | 83.67 ± 0.86          |
> | NTGA                                                    | 82.99 ± 0.40          |
> | Adversarial Poisoning                                   | **77.35 ± 0.43**      |
> | REM [Ref_1]                                             | 85.63 ± 1.05          |
>
> In short, while REM [Ref_1] achieved SOTA when $\rho=\epsilon/2=4/255$, its effectiveness as a delusive attack is inferior when $\rho=\epsilon=8/255$. (By the way, this result cannot be found in [Ref_1], since they did not report the standard accuracy for the case of $\rho=\epsilon$.)
>
> ---
>
> > Thus, it is unclear that this paper made the fair comparison between current SOTA poisoning attacks on adversarial trained network.
>
> This is a very good question. The threat considered in this work is to degrade the robust accuracy, which is different from the goal of the existing works on delusive attacks. To reflect the difference, we call our threat stability attack.
>
> Below we focus on a fair comparison between our method and REM [Ref_1]. Results of other existing methods can be found in Table 2 of the submission. For a fair comparison, all poisons are bounded by $\epsilon=8/255$, and conventionally the defense is adversarial training with $\rho=8/255$. Since the robust accuracy is NP-hard to compute [Ref_9], it is common practice to use test-time perturbations as a proxy for evaluating test robustness. To this end, we use AutoAttack (a reliable evaluation metric via an ensemble of diverse attacks) to evaluate the robust accuracy. For a fair comparison, here we apply a very simple trick called EOT in our method, since the trick is also used by REM [Ref_1]. Results are reported as follows:
>
> | Stability attacks (aimed at degrading robust accuracy) | Time cost (hours) | Robust accuracy (%) |
> | ------------------------------------------------------ | ----------------- | ------------------- |
> | REM [Ref_1]                                            | 23                | 33.09 ± 0.24        |
> | Hypocritical Perturbation (ours)                       | **0.5**           | **32.79 ± 0.37**    |
>
> We make the following observations:
>
> - In a fair comparison, **our method outperforms REM in degrading robust accuracy**.
> - Importantly, **our method is significantly more efficient than REM**. The time cost of REM is nearly 50 times that of us!
> - Interestingly, we find that REM can be considered as an effective stability attack, though it was originally proposed as a delusive attack. It is worth noting that the robust accuracy was not evaluated in [Ref_1]. In this sense, the effectiveness of REM as an stability attack can be regarded as one of our novel findings.

---

> ### Author Response · Authors · 2022-08-02
> **Response to Reviewer J69n (Part 1)**
>
> Thanks a lot for your insightful feedback. We address your concerns below, but let us know if any parts remain unclear.
>
> ---
>
> > Low originality: Missing comparison with the recent key reference [Ref_1]
>
> Thanks for your comments. Actually, we cited and discussed [Ref_1] in the submission. The detailed discussion and experimental comparison were deferred to Appendix A due to space limitation. Appendix A can be found in the Supplementary Material, where similarities, dissimilarities, and experimental results were presented.
>
> We would like to emphasize that **the motivation and conclusions of this work are completely different from those of [Ref_1]**.
>
> Evidently, both this work and [Ref_1] break through the defense of adversarial training. The point, however, is that adversarial training has two *distinct* defense abilities:
>
> 1. Adversarial training is capable of improving model robustness against test-time adversarial examples [Ref_4].
> 2. Adversarial training is capable of protecting the model's standard accuracy from delusive attacks (a type of data poisoning) [Ref_5].
>
> **This work aims to hinder the first ability of adversarial training** by allowing the attacker to perturb the training data (we introduce the term "stability attacks" to refer to this malicious task):
>
> - Our main conclusion is that when the training data is perturbed by a budget $\epsilon$, adversarial training with the budget $\epsilon$ can be broken, as it fails to provide *test robustness* against $\epsilon$-bounded adversarial examples.
> - For example, we find that on CIFAR-10 when the training data is poisoned by a budget $\epsilon=8/255$, the *robust accuracy* can be degraded from 46.99% to 35.44%. Note that the models here are produced by adversarial training with the budget $\epsilon=8/255$.
>
> In contrast, **the goal of [Ref_1] is to hinder the second ability of adversarial training**:
>
> - Their core argument is that when the training data is perturbed by a budget $\epsilon$, adversarial training with a budget $\rho$ **(notably, they require $\rho < \epsilon$)** can be broken, in terms of failing to protect the *standard accuracy*.
> - For example, they observed that on CIFAR-10 when the training data is poisoned by a budget $\epsilon=8/255$, the *standard accuracy* can be degraded from 89.51% to 47.51%. Note that their models were produced by adversarial training with a budget $\rho=4/255$.
>
> The delusive attack method proposed in [Ref_1] looks promising, because they kept the budget $\rho$ much lower than the budget $\epsilon$ in their experiments. However, it is well known that adversarial training can effectively mitigate the success of delusive attacks when $\rho=\epsilon$ [Ref_5] [Ref_6] [Ref_7] [Ref_8]. In other words, the second ability of adversarial training is largely unbreakable when $\rho=\epsilon$ [Ref_5]. On the contrary, this work shows that the first ability of adversarial training can still be broken by perturbing the training data when $\rho=\epsilon$. This is an important contribution that distinguishes this work from [Ref_1] and other previous works.
>
> ---
>
> > The attack generation algorithm of training-time and test-time perturbation is similar to [Ref_1].
>
> Thanks for this question. First, [Ref_1] did not apply test-time perturbations.
>
> - Only standard accuracy, not robust accuracy, was reported in [Ref_1].
> - On the contrary, the main focus of this work is the robust accuracy.
>
> Second, the poisoning method proposed in [Ref_1] is much more complicated than that used in this work.
>
> - Specifically, [Ref_1] generated their poisons by solving the following **bi-level** objective function:
>   &emsp;&emsp;&emsp;&emsp;&emsp;&emsp;&emsp;&emsp;$\min_{\|p_i\|\le\epsilon} \max_{\|\delta_i\|\le\rho}\ell(f_{\theta^*}(x_i+p_i+\delta_i),y_i),$
>   where the noise generator $f_{\theta^*}$ is pre-trained by solving the following **three-level** optimization process (Equation (4) in [Ref_1]):
>   &emsp;&emsp;&emsp;&emsp;&emsp;&emsp;&emsp;&emsp;$\theta^* = \arg\min_{\theta}\sum_{i=1}^n \min_{\|p_i\|\le\epsilon} \max_{\|\delta_i\|\le\rho} \ell(f_{\theta}(x_i+p_i+\delta_i), y_i).$
>
> - In contrast, our poisons are crafted via the following **single-level** objective function (this objective simply follows Equation (14) in [Ref_5]):
>   &emsp;&emsp;&emsp;&emsp;&emsp;&emsp;&emsp;&emsp;$\min_{\|p_i\|\le\epsilon}\ell(f_{craft}(x_i+p_i), y_i),$
>   where our crafting model $f_{craft}$ is obtained with adversarial training (a **bi-level** optimization process):
>   &emsp;&emsp;&emsp;&emsp;&emsp;&emsp;&emsp;&emsp;$f_{craft} = \arg\min_{f} \sum_{i=1}^n \max_{\|p_i\|\le\epsilon/4}\ell(f(x_i+p_i), y_i).$
>
> Clearly, **our method is much simpler than [Ref_1]**. This simplicity gives our method a significant advantage in efficiency. For example, **the time cost of [Ref_1] to generate poison for CIFAR-10 is about 23 hours, while our method only takes 0.5 hours**. The time cost here is evaluated on a single NVIDIA GeForce RTX 3090 GPU.

---

> ### Author Response · Authors · 2022-08-08
> **Thank you again for the review and hope the reviewer can check out our response**
>
> Dear Reviewer J69n,
>
> Thank you again for your detailed review. As the discussion period is closing soon, we hope the reviewer can take a look at our response and reevaluate our paper based on the detailed response along with the revised paper.
>
> In our response, we clarified the differences between this work and prior work. We also answered your comments on non-robust features. In the revised paper we performed an additional feature-level analysis to empirically validate the relationship with non-robust features.
>
> Let us know if you have further questions about our paper and we look forward to hearing from you.
>
> Sincerely,
> Paper11926 Authors

---

### Official Review · Reviewer_dZbx · 2022-07-11

**Rating:** 6
**Confidence:** 3
**Soundness:** 3 good
**Presentation:** 4 excellent
**Contribution:** 3 good

**Summary:**

This paper presents sability attack against the conventional adversarial training process, aiming to reduce the eventual robust accuracy of the resulting model. Specifically, the corresponding hypocritical perturbations are applied into training data as a training-time attack. Theoretical analysis is provided to support the idea of hypocritical perturbations. Experimental results on commonly used classification datasets like CIFAR-10 demonstrate the effeciveness of the proposed method.

**Questions:**

My questions are summarized in the weaknesses.

**Limitations:**

Limitations are discussed as pointed out in the checklist.

**Strengths And Weaknesses:**

# Strengths

1. This paper is technically sound and clear around the theoretical analysis. Experimental results are significant, which well support the theory.

2. The writing quality of the paper is good overall. Specifically, the background of problem is smoothly introduced.

3. The proposed method is sufficiently evaluated. To be specific, key attacks like FGSM, PGD, CW, and AutoAttack are present for robustness evaluation. Meanwhile, the experiments cover four datasets, namely CIFAR-10/100, SVHN, and Tiny-ImageNet, which are sufficient to demonstrate the effectiveness for the proposed method.

4. The proposed method successfully compromised adversarial training methods. However, countermeasure (adaptive defense) is also proposed and evaluated.

# Weaknesses

1. Overall this is a good paper and I did not find many problems.

---

> ### Author Response · Authors · 2022-08-02
> **Response to Reviewer dZbx**
>
> Thank you very much for your appreciation and kind words.

---

### Official Review · Reviewer_bFP2 · 2022-07-11

**Rating:** 5
**Confidence:** 3
**Soundness:** 3 good
**Presentation:** 3 good
**Contribution:** 2 fair

**Summary:**

This paper propose a new threat model called stability attack. The goal of stability attack is to hinder model from being robust to adversarial attacks. The author  proposes hypocritical perturbation as a method for stability attack and shows that hypocritical perturbation is harmful in terms of adversarial robustness in a simple gaussian mixture setting. Finally, the author shows that 2 adversarial training is enough for protecting stability attack.

**Questions:**

I have no question.

**Limitations:**

- As mentioned in the manuscript, the standard accuracy of a robustly leanred prediction model based on training data poisoned by the stability attack is  better than the standard accuracy of a robustly trained model based on training data poisoned by other methods. In this sense, the stability attack is less serious than other poisoned methods which degrade the standard accuracy as well as the robust accuracy simultaeously.

**Strengths And Weaknesses:**


Pros:
- As far as I know, this is the first work that studies the robustness of adversarial training against train data poisoning called stability attack.
- As [1] noted, hypocritical perturbation is known to be a weak attack in terms of the standard accuracy, but this paper illustrates that it can be a strong attack in terms of the robust accuracy.
- The intuition of stability attack for vulnerability of adversarial attack is well demonstrated.
Cons:
- It seems that hypocritical perturbation is main content, but is not well described. Especially, the comparison of adversarial perturbation and hypocritical perturbation lacks.
- It seems that Stability Attack in Table 2 and Adversarial Poisoning and Hyp. and Adv. in Table (3, 4) are identical, respectively. The consistency of naming the attacks is necessary.
- The description of the poisoning methods in Table 2 is not given.
- In Table 3,  Adv. and Hyp are only considered for the methods of data poisoning. It would be more appropriate to do evaluation against the various attacks as in Table 2.
- The final message of the paper is that by using a larger perturbation size, the hypocritical perturbation can be defended. However, this message would be obvious and so the novelty would not be high.

[1] Lue Tao, Lei Feng, Jinfeng Yi, Sheng-Jun Huang, and Songcan Chen. Better safe than sorry: Preventing delusive adversaries with adversarial training. In NeurIPS, 2021.

---

> ### Author Response · Authors · 2022-08-02
> **Response to Reviewer bFP2 (Part 2)**
>
> > As mentioned in the manuscript, the standard accuracy of a robustly leanred prediction model based on training data poisoned by the stability attack is better than the standard accuracy of a robustly trained model based on training data poisoned by other methods. In this sense, the stability attack is less serious than other poisoned methods which degrade the standard accuracy as well as the robust accuracy simultaeously.
>
> This is a great question. In this work, we focus on degrading the robust accuracy, so we do not impose additional restrictions on the impact of stability attacks on the standard accuracy. Having that said, we would like to point out that increasing the standard accuracy is more *insidious* than decreasing it. For example, if the poisoned model exhibits higher standard accuracy, practitioners would be more easily enticed to deploy it in a real-world system. However, as its robust accuracy is actually undesirably low, the system is prone to losing its normal function when encountering test-time perturbations. In short, the negative impacts of stability attacks would be very serious. Thus, it is imperative to design better defense methods to mitigate the threat of stability attacks.
>
> **[Update: 2022.08.04]** Thanks for this question. To explain the rationality of higher natural accuracy, we have added this discussion in Appendix G of the updated manuscript.
>
> ---
>
> **References**
>
> [1] Yuan, C. H., & Wu, S. H. (2021, July). Neural tangent generalization attacks. In *International Conference on Machine Learning* (pp. 12230-12240). PMLR.
> [2] Fowl, L., Goldblum, M., Chiang, P. Y., Geiping, J., Czaja, W., & Goldstein, T. (2021). Adversarial Examples Make Strong Poisons. Advances in Neural Information Processing Systems, 34, 30339-30351.
> [3] Yu, D., Zhang, H., Chen, W., Yin, J., & Liu, T. Y. (2021). Indiscriminate poisoning attacks are shortcuts. arXiv preprint arXiv:2111.00898.
> [4] Schmidt, L., Santurkar, S., Tsipras, D., Talwar, K., & Madry, A. (2018). Adversarially robust generalization requires more data. Advances in neural information processing systems, 31.
> [5] Shaeiri, A., Nobahari, R., & Rohban, M. H. (2020). Towards deep learning models resistant to large perturbations. arXiv preprint arXiv:2003.13370.

---

> ### Author Response · Authors · 2022-08-02
> **Response to Reviewer bFP2 (Part 1)**
>
> Thank you very much for the constructive feedback. We address your concerns below, but let us know if any parts remain unclear.
>
> ---
>
> > It seems that hypocritical perturbation is main content, but is not well described. Especially, the comparison of adversarial perturbation and hypocritical perturbation lacks.
>
> Thanks for your suggestion. Here we make a brief comparison.
>
> The hypocritical perturbation is optimized to minimize the loss of a pre-trained crafting model. Formally, $\min_{\|p_i\|\le\epsilon}\ell(f_{craft}(x_i+p_i), y_i).$
>
> In contrast, the adversarial perturbation is optimized to maximize the loss of the crafting model. Formally, $\max_{\|p_i\|\le\epsilon}\ell(f_{craft}(x_i+p_i), y_i).$
>
> Following your suggestion, we have added a more detailed description of the hypocritical perturbation, the adversarial perturbation, and their origins in Section 2.1 (Preliminaries) of the updated manuscript.
>
> ---
>
> > It seems that Stability Attack in Table 2 and Adversarial Poisoning and Hyp. and Adv. in Table (3, 4) are identical, respectively. The consistency of naming the attacks is necessary.
>
> Thanks for your careful observation. We have corrected this naming issue in the updated manuscript. Specifically, we make the following changes:
>
> - In Table 2: "Stability Attacks" -> "Hypocritical Perturbation"
> - In Table 8: "Stability Attacks" -> "Hypocritical Perturbation"
> - In Table 11: "Hypocritical Features" -> "Hypocritical Perturbation"
>
> ---
>
> > The description of the poisoning methods in Table 2 is not given.
>
> Thanks for pointing this out. We have added a description of the poisoning methods in Section 5 (Experiments) of the updated manuscript.
>
> ---
>
> > In Table 3, Adv. and Hyp are only considered for the methods of data poisoning. It would be more appropriate to do evaluation against the various attacks as in Table 2.
>
> Thanks for this question. Several previous works are not compared in Table 3 because in their official code repositories, the generated poisons (or scripts to generate poisons) for the datasets in Table 3 are not provided. For example, the code of [1] supports the CIFAR-10 dataset and a two-class subset of ImageNet, while the code of [2] supports the CIFAR-10 and the 1000-class ImageNet datasets. Fortunately, all previous works support CIFAR-10. Thus, it is reasonable to compare their methods on CIFAR-10, while covering other datasets (such as SVHN, CIFAR-100, and Tiny-ImageNet) to further demonstrate the effectiveness of the proposed method, as done in [2] and [3]. So does this work.
>
> ---
>
> > The final message of the paper is that by using a larger perturbation size, the hypocritical perturbation can be defended. However, this message would be obvious and so the novelty would not be high.
>
> Thanks for this question. The message from the adaptive defense (by enlarging the defense budget) has both good and bad aspects. The good news is that, as the reviewer has pointed out, the adaptive defense largely prevents the hypocritical perturbation. However, it also has some *potential* negative effects, which are summarized as follows:
>
> - The adaptive defense improves robust accuracy at the price of clean accuracy.
> - It is time-consuming to find an appropriate defense budget for adversarial training using grid search on real-world datasets.
> - Adversarial training with large budgets may lead to other learning obstacles such as inherent large sample complexity [4] and optimization difficulties [5].
>
> These limitations indicate that the current defense is still unsatisfactory. We have added a description of these limitations in Section 5 (Experiments) of the updated manuscript. In summary, the proposed method cannot be well defended by the adaptive defense. This work is only a small step towards eliminating the impacts of the hypocritical perturbation, and more efforts are required to achieve complete robustness against stability attacks.
>
> In addition, the paper also attempts to convey two other *novel* messages:
>
> 1. The threat model of stability attacks introduced in this work has never been studied before.
> 2. The hypocritical perturbation can be a strong attack in terms of robust accuracy, whereas previous work considered it to be a weak attack in terms of standard accuracy.
>
> These messages are also recognized by the reviewer. All in all, we believe that the novelty of the paper is enough.

---

> ### Author Response · Authors · 2022-08-08
> **Thank you again for the review and hope the reviewer can check out our response**
>
> Dear Reviewer bFP2,
>
> Thank you again for your detailed review. As the discussion period is closing soon, we hope the reviewer can take a look at our response and reevaluate our paper based on the detailed response along with the revised paper.
>
> In our response, we illustrated the limitations of the current defense. We also answered each of your minor comments. In the revised paper we added a more detailed description of the hypocritical perturbation, and explained the rationale for higher natural accuracy of the hypocritical perturbation.
>
> Let us know if you have further questions about our paper and we look forward to hearing from you.
>
> Sincerely,
> Paper11926 Authors

---

### Author Response · Authors · 2022-08-09
**Summary of Changes in Revision**

We again thank all reviewers for their valuable feedback.

We updated our submission with the following key changes:

- [Section 2.1] We added a more detailed description of the origins of hypocritical perturbations (Reviewer bFP2 and Reviewer UnFX).
- [Section 5] We added a description of compared methods (Reviewer bFP2), illustrated several limitations of the adaptive defense (Reviewer bFP2), and replaced several expressions "stability attacks" with "hypocritical perturbation" to avoid ambiguity (Reviewer bFP2 and Reviewer J69n).
- [Appendix A] We revised this section for a fair comparison with previous work, showing the superiority of the proposed method in both effectiveness and efficiency (Reviewer J69n and Reviewer UnFX).
- [Appendix E] We presented a feature-level analysis to provide an additional empirical evidence on the role of non-robust features in the success of the hypocritical perturbation (Reviewer J69n).
- [Appendix F] We included a statement of the broader impact of this work, including potential real-world consequences (Reviewer UnFX).
- [Appendix G] We added a detailed discussion on the trade-off between the clean and robust accuracy (Reviewer bFP2, Reviewer J69n, and Reviewer UnFX).

Furthermore, we have now uploaded the full submission (main paper + the appendix) as a single pdf file.

---

### Meta-Review · Area_Chair_8Pa8 · 2022-08-29

**Recommendation:** Accept
**Confidence:** Certain

**Metareview:**

This paper proposes a new threat model called stability attack, which aims to hinder model from being robust to adversarial attacks. The author proposes hypocritical perturbation as a method for stability attack and shows that hypocritical perturbation can indeed decrease the adversarial robustness of a model trained in a simple gaussian mixture setting. The reviewers agree that the problem being studied is interesting, the proposed method is well motivated, and the experiments are mostly convincing. The authors are encouraged to merge the new results during the rebuttal into the publication and discuss more on the efficiency of the proposed method.

**Award:**

No

---

### Decision · Program_Chairs · 2022-09-14

Accept